# Thermodynamics of diamond formation from hydrocarbon mixtures in planets

Bingqing Cheng [1], Sebastien Hamel [2] & Mandy Bethkenhagen [1,3]

Hydrocarbon mixtures are extremely abundant in the Universe, and diamond formation from them can play a crucial role in shaping the interior structure and evolution of planets. With first-principles accuracy, we first estimate the melting line of diamond, and then reveal the nature of chemical bonding in hydrocarbons at extreme conditions. We finally establish the pressure-temperature phase boundary where it is thermodynamically possible for diamond to form from hydrocarbon mixtures with different atomic fractions of carbon. Notably, here we show a depletion zone at pressures above 200 GPa and temperatures below 3000 K-3500 K where diamond formation is thermodynamically favorable regardless of the carbon atomic fraction, due to a phase separation mechanism. The cooler condition of the interior of Neptune compared to Uranus means that the former is much more likely to contain the depletion zone. Our findings can help explain the dichotomy of the two ice giants manifested by the low luminosity of Uranus, and lead to a better understanding of (exo-)planetary formation and evolution.

Carbon and hydrogen are the fourth and the most abundant elements in the Universe[1], and their mixture is the simplest basis to form organic compounds. In our Solar System, the Cassini mission revealed lakes and seas of liquid hydrocarbons on the surface of Titan[2], and the New Horizon spacecraft detected methane frost on the mountains of Pluto[3]. In Neptune and Uranus, methane is a major constituent with a measured carbon concentration from around 2% in the atmosphere[4] and a concentration up to (assumed) 8% in the interior[5]. The methane in the atmosphere absorbs red light and reflects blue light, giving the ice giants their blue hues[6]. Moreover, numerous recently discovered extrasolar planets, some orbiting carbon-rich stars, have spurred a renewed interest in the high-pressure and high-temperature behaviors of hydrocarbons[7].

Diamond formation from C/H mixtures is particularly relevant; the "diamonds in the sky" hypothesis[8] suggests that diamonds can form in the mantles of Uranus and Neptune. The diamond formation and the accompanying heat release may explain the long-standing puzzle that Neptune (but not Uranus) radiates much more energy than that it receives from the Sun[9]. Diamond is dense and will gravitate into the core of the ice giants. For white dwarfs, Tremblay et al.[10]

interpreted the crystallization of the carbon-rich cores to influence the cooling rate.

Many experimental studies have probed the diamond formation from C/H mixtures, but the experiments are extraordinarily challenging to perform and interpret because of the extreme thermodynamic conditions, kinetics, chemical inhomogeneities, possible surface effects from the sample containers, and the need to prove diamond formation inside a diamond anvil cell (DAC). Three DAC studies on methane disagree on the temperature range: Benedetti et al. reported diamond formation between 10 to 50 GPa and temperatures of about 2000 K to 3000 K[11]; between 10 to 80 GPa, Hirai et al. reported diamond formation above 3000 K[12]; while Lobanov et al. reported the observation of elementary carbon at about 1200 K, and a mixture of solid carbon, hydrogen, and heavier hydrocarbons at above 1500 K[13]. In methane hydrates, Kadobayashi et al. reported diamond formation in a DAC between 13 and 45 GPa above 1600 K but not at lower temperatures[14]. Laser shock-compression experiments found diamond formation in epoxy (C,H,Cl,N,O)[15] and polystyrene (-$C_8H_8$-)[16], but none in polyethylene (-$C_2H_4$-)[17].

[1]The Institute of Science and Technology Austria, Am Campus 1, 3400 Klosterneuburg, Austria. [2]Lawrence Livermore National Laboratory, Livermore, CA 94550, USA. [3]École Normale Supérieure de Lyon, Université Lyon 1, Laboratoire de Géologie de Lyon, CNRS UMR 5276, 69364 Lyon, Cedex 07, France. ✉e-mail: bingqing.cheng@ist.ac.at

Moreover, there is a mismatch between the experimental results and theoretical predictions particularly regarding the pressure range of diamond formation. Density functional theory (DFT) combined with crystal structure searches at the static lattice level predicted that diamond and hydrogen are stable at pressures above about 300 GPa[18,19], while hydrocarbon crystals are stable at lower pressures[18–22]. Based on DFT molecular dynamics (MD) simulations of methane, Ancilotto et al. concluded that methane dissociates into a mixture of hydrocarbons below 100 GPa and is more prone to form diamond at above 300 GPa [23], Sherman et al. classified the system into stable methane molecules (<3000 K), a polymeric state consisting of long hydrocarbon chains (4000-5000 K, 40–200 GPa), and a plasma state (>6000 K) [24]. However, these simulations are constrained to small system sizes and short time scales, so that it is impossible to distinguish between the formation of long hydrocarbon chains and the early stage of diamond nucleation. Using a semiempirical carbon model, Ghiringhelli et al.[25] determined that the diamond nucleation rate in pure liquid carbon is rapid at 85 GPa, 5000 K but negligibly small at 30 GPa, 3750 K, and then extrapolated the nucleation rate to mixtures employing an ideal solution model.

In this work, we go beyond the standard first-principles methods, and study the thermodynamics of diamond formation in C/H mixtures, by constructing and utilizing machine learning potentials (MLPs) trained on DFT data. To the best of our knowledge, this is the first MLP fitted for high-pressure mixtures, and the only one available for C/H mixtures with arbitrary compositions and applicable from low $P$-$T$ conditions to about 8000 K and 800 GPa. We first quantitatively estimate the coexistence line between diamond and pure liquid carbon at planetary conditions. We then reveal the nature of the chemical bonds in C/H mixtures at high-pressure high-temperature conditions. Finally, we determine the thermodynamic driving force of diamond formation in C/H mixtures, taking into account both the ideal and the non-ideal effects of mixing. We thereby establish the phase boundary where diamond can possibly form from C/H mixtures at different atomic fractions and $P$-$T$ conditions.

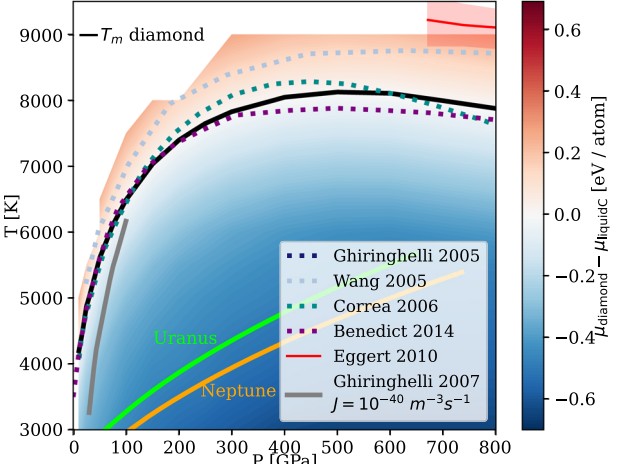

**Fig. 1 | Thermodynamics for diamond formation in pure liquid carbon.** The color scale shows the chemical potential of diamond, $\Delta\mu_D$, referenced to pure liquid carbon. The stability region of graphite at $P \lesssim 10$ GPa is not shown. The melting curve $T_m$ is compared to previous calculations using thermodynamic integration (TI) employing a semi-empirical potential by Ghiringhelli et al.[31], TI using DFT by Wang et al.[28], coexistence DFT simulations by Correa et al.[29], an analytic free-energy model fitted to DFT data by Benedict et al.[30], and a shock-compression experiment by Eggert et al.[27] with uncertainties indicated using the shaded area. The gray line shows the inferred threshold conditions with a diamond nucleation rate $J$ of $10^{-40}$m$^{-3}$s$^{-1}$ from pure liquid carbon by Ghiringhelli et al.[25]. The $P$-$T$ curves of planetary interior conditions for Uranus (green line) and Neptune (orange line) are from Ref. [40].

## Results

### Diamond formation in pure liquid carbon

Although planets or stars typically contain a low percentage of carbon[4], it is useful to start with a hypothetical environment of pure carbon. This is to establish the melting line of diamond and to facilitate the subsequent analysis based on C/H mixtures. Moreover, the high-pressure carbon system has experimental relevance in diamond synthesis and Inertial Confinement Fusion applications[26].

Figure 1 shows the chemical potential difference $\Delta\mu_D \equiv \mu_{\text{diamond}} - \mu_{\text{liquidC}}$ between the diamond and the pure liquid carbon phases calculated using our MLP at a wide range of pressures and temperatures. Our calculated melting line $T_m$ of diamond in pure liquid carbon (solid black curve) is compared to other theoretical work and experimental shock-compression data (Fig. 1a). Our $T_m$ is re-entrant at above 500 GPa, because liquid carbon is denser than diamond at higher pressures. This shape has been observed for the experimental melting line[27]. It was previously predicted using DFT simulations on smaller systems[28–30], but not captured in the free energy calculations performed using a semi-empirical LCBOP carbon model[31].

Although diamond solidification is thermodynamically favorable below the melting line, undercooled liquids can remain metastable for a long time as solidification is initiated by a kinetically activated nucleation process[32]. The only previous study that has quantified the diamond nucleation rate is by Ghiringhelli et al.[25] using the LCBOP carbon model: the threshold $J = 10^{-40}$m$^{-3}$s$^{-1}$ is indicated by the gray line in Fig. 1, and above this line the diamond formation rate is negligible even at the celestial scale. Overall, we find that the pure carbon system is deeply undercooled at the $P$-$T$ conditions in the two icy planets (green and orange lines in Fig. 1).

### The nature of C-H bonds

Going beyond the pure carbon case, we investigate the nature of the chemical bonds in C/H mixtures at conditions relevant for planetary interiors. The high-pressure behavior of hydrocarbons is also crucial in many shock-compression experiments for the development of fusion energy platforms and Inertial Confinement Fusion capsules[33]. The properties of the covalent C-C and C-H bonds are well-known at ambient conditions, but it is unclear how extreme conditions affect these bonds. DFT studies coupled with harmonic approximations have predicted a variety of hydrocarbon crystals to be stable at $P \leq 300$ GPa[18–22], but these studies are restricted to low temperatures as the melting lines of hydrogen and methane are below 1000 K and 2000 K[12,34], respectively, while harmonic approximations break down completely for these liquids.

We performed MD simulations using our dissociable MLP for C/H mixtures over a wide range of thermodynamic conditions. We focus on the CH$_4$ composition to directly compare to previous studies. Other compositions can be analyzed in the same way and yield qualitatively similar behaviors. At $T < 2500$ K, the MD is not ergodic within the simulation time of 100 ps, and therefore analysis is performed only at temperatures above this threshold. Figure 2a shows the snapshots of carbon bonds from the MD simulations of the CH$_4$ system. At 4000 K and $P = 100$ GPa, 200 GPa, and 600 GPa, the system is primarily composed of various types of hydrocarbon chains. The formation of longer chains at higher pressures is consistent with the observations in previous DFT MD studies[23,24], although the DFT simulations have severe finite size effects because polymer chains consisting of just a few carbon atoms can connect with their periodic images and become infinitely long. At high pressures, the chains assemble carbon networks, and the system shows more obvious signs of spatial inhomogeneity of carbon atoms.

In our chemical bond analysis, a C-C bond is identified whenever the distance between a pair of carbon atoms is within 1.6 Å, and a C-H bond is defined using a cutoff of 1.14 Å. The cutoffs are larger than the typical bond lengths to eliminate the misidentification of broken

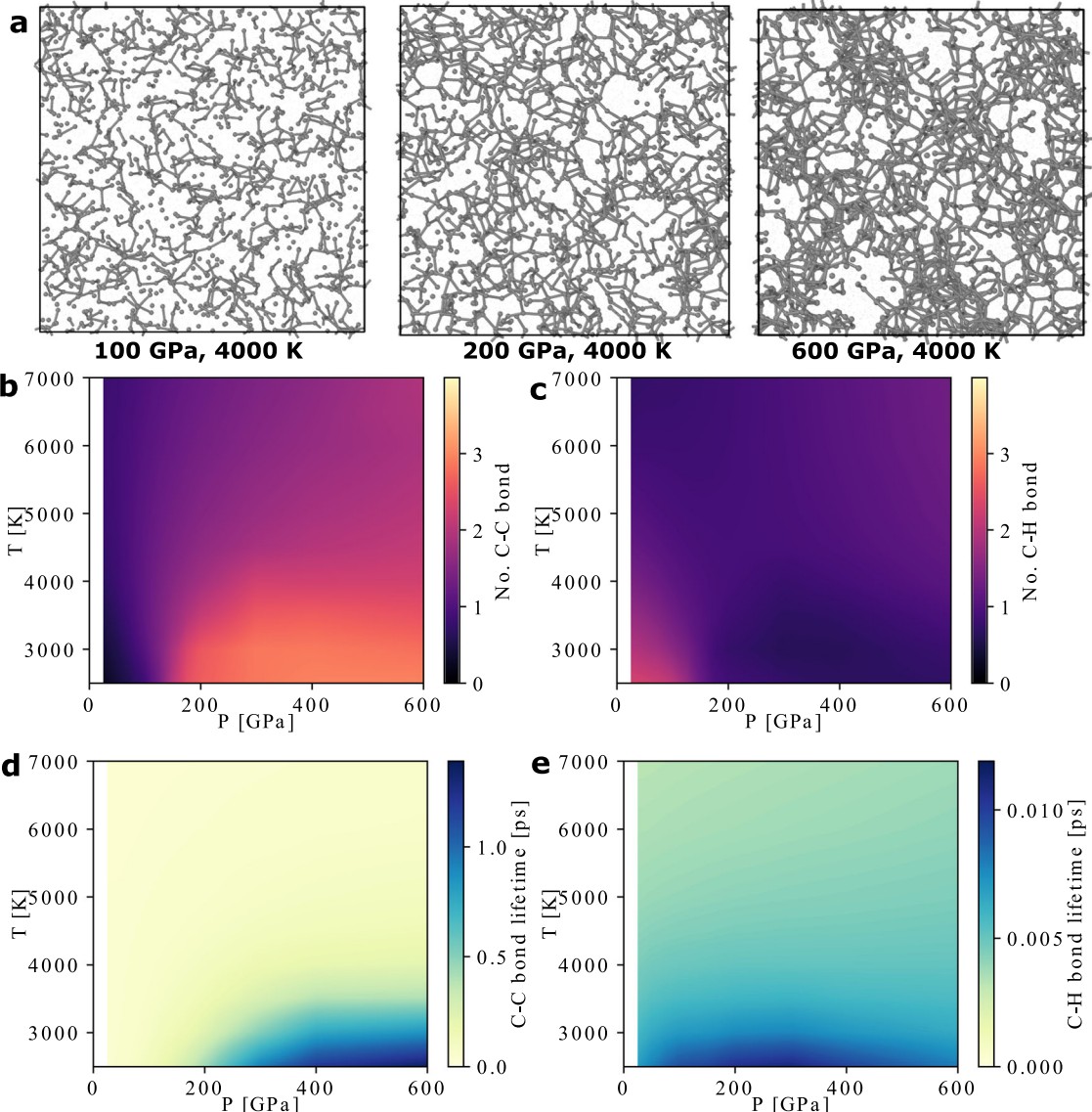

**Fig. 2 | Bonding behavior in the CH₄ mixture at high-pressure high-temperature conditions. a** Snapshots from MD simulations using the MLP. Carbon atoms are shown as small gray spheres, while hydrogen atoms are not drawn for clarity. Bonds are drawn for C-C pairs with distances within 1.6 Å. **b,c** Average number of C-C bonds (**b**) and C-H bonds (**c**) per carbon atom from MD simulations of CH₄ composition. **d,e** Average lifetimes of the C-C bonds (**d**) and C-H bonds (**e**).

bonds due to thermal fluctuations. The average number of C-C and C-H bonds at different conditions are shown in Fig. 2b, c. The number of bonds varies smoothly as a function of *P* and *T*.

The average number of the C-C bonds decreases with temperature. Moreover, as illustrated in the Supplementary Information, at a certain condition the carbon atoms in the system have varying number of C-C bonds, rather than all having the same number of bonds. These suggest that at $T \geq 2500$ K the system is not made of hydrocarbon crystals that were predicted to be stable at low temperatures in previous DFT studies[18–22]. The average number of C-H bonds for each carbon atom is close to one at all conditions considered here even though the overall composition is CH₄, indicating that most hydrogen atoms are not bonded to any carbons.

To determine the lifetimes of the C-C and the C-H bonds, we recorded the time it takes for a newly formed bond to dissociate during the MD simulations. Figure 2d, e show the average bond lifetimes. The C-H bond lifetimes are extremely short, less than about 0.01 ps. The C-C bonds are more long-lived, yet only have a mean lifetime of less than about 1 ps at all the conditions considered here. Such short lifetimes are consistent with previous DFT MD simulations of CH₄[24].

The short bond lifetimes indicate that the hydrocarbon chains in the systems decompose and form quickly. In other words, the C/H mixture behaves like a liquid with transient C-C and C-H bonds.

## Thermodynamics of C/H mixtures

We then determine the chemical potentials of carbon in C/H mixtures, $\Delta\mu^C(\chi_C)$, as a function of the atomic fraction of carbon, $\chi_C = N_C/(N_C + N_H)$. This, combined with the chemical potential difference $\Delta\mu_D$ between diamond and pure carbon liquid, establishes the thermodynamic phase boundary for diamond formation from C/H mixtures with varying atomic ratios.

Dilution will usually lower the chemical potential of carbon in a mixture, which can be understood from the ideal solution assumption: $\mu_{id}^C = k_B T \ln(\chi_C)$. However, the ideal solution model neglects the atomic interactions. To consider non-ideal mixing effects, we compute the chemical potentials of mixtures using the MLP. This is not an easy task, because traditional particle insertion methods[35] fail for this dense liquid system, and thermodynamic integration from an ideal gas state to the real mixture[36] is not compatible with the MLP. We employ the newly developed S0 method which accounts for both the ideal and

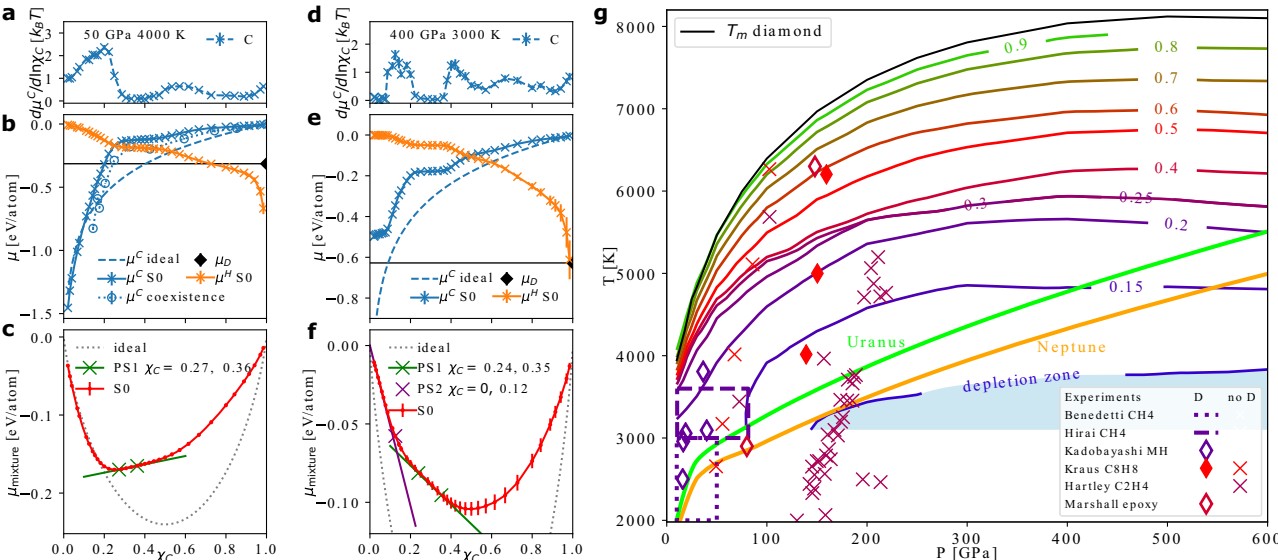

**Fig. 3 | Thermodynamic behaviors of C/H mixtures. a,d** $d\mu^C/d\ln(\chi_C)$ computed from static structure factors using Eqn. (1), at 50 GPa, 4000 K (**a**) and at 400 GPa, 3000 K (**d**). **b,e** The chemical potential per carbon atom ($\mu^C$) in C/H mixtures, estimated using the S0 method (blue crosses connected by solid curve), the coexistence method (blue hollow symbols connected by dotted curve), and the ideal solution approximation (blue dashed curve). The chemical potential per hydrogen atom ($\mu^H$) is shown as orange line with crosses. The chemical potential of diamond ($\mu_D$) is indicated by the black diamond symbol and the black horizontal line. **c,f** The free energy per atom of C/H mixtures with different carbon fractions $\chi_C$. The free energy of the ideal solution is shown as dashed gray curve, and the result from the S0 method is shown in red. The green line illustrates the constructed double tangent, and the two green crosses show the composition of the liquids forming after phase separation (PS1). The purple line is a tangent and the purple cross shows the composition of the more carbon-rich phase after PS2. The standard mean errors are indicated using the error bars in **a**-**f**. **g** The conditions below which diamond formation is thermodynamically possible at different carbon ratios (colored solid lines). The shaded area below the bright blue line indicates the depletion zone, where diamond formation is thermodynamically favorable regardless of the carbon ratio. Experimental observation of diamonds (D) is indicated by diamond symbols and dashed/dotted rectangular regions, while no diamond observation (no D) is marked by cross symbols: laser-heated DAC data for methane [11,12], for methane hydrate (MH) [14], and shock-compression experiments for polystyrene (-$C_8H_8$-) [16], polyethylene (-$C_2H_4$-) [17], and epoxy [15]. The symbols are colored according to the $N_C/(N_C + N_H)$ ratio of the starting materials, using the same color scheme for the phase boundaries. The *P-T* curves of planetary interior conditions for Uranus (green line) and Neptune (orange line) are from Ref. [40].

the non-ideal contributions to the chemical potentials [37]:

$$\left(\frac{d\mu^C}{d\ln\chi_C}\right)_{T,P} = \frac{k_B T}{(1-\chi_C)S_{CC}^0 + \chi_C S_{HH}^0 - 2\sqrt{\chi_C(1-\chi_C)}S_{CH}^0}, \quad (1)$$

where $S_{CC}^0$, $S_{CH}^0$, and $S_{HH}^0$ are the values of the static structure factor between the said types of atoms at the limit of infinite wavelength [37], which can be determined from equilibrium MD simulations of a C/H mixture with a given carbon fraction $\chi_C$. $\mu^H$ is then fixed using the Gibbs-Duhem equation. Note that only the relative chemical potential is physically meaningful, and we conveniently select the reference states to be the pure carbon and hydrogen liquids, i.e. $\mu^C(\chi_C = 1) = 0$ and $\mu^H(\chi_C = 0) = 0$. We obtained $\mu^C$ and $\mu^H$ at different $\chi_C$ on a grid of *P-T* conditions between 10 GPa–600 GPa and 3000 K–8000 K, by numerically integrating $d\mu^C/d\ln\chi_C$.

$d\mu^C/d\ln\chi_C$ at *P* = 50 GPa, *T* = 4000 K and *P* = 400 GPa, *T* = 3000 K are shown in Fig. 3a, d, respectively. For both sets, these values deviate from the ideal behavior (i.e. constant at 1), and have maxima and minima around certain compositions. The corresponding chemical potentials are plotted in Fig. 3b, e, while the results at other conditions are shown in Fig. 4 of the Methods. As an independent validation, we also computed $\mu^C$ using the coexistence method described in the Methods, although this approach is in general less efficient and can become prohibitive if carbon concentration or diffusivity is low. The values from the coexistence method are shown as the hollow symbols in Fig. 3b, in agreement with the S0 method. As the statistical accuracy of the S0 method is much better compared to the coexistence approach, all the subsequent analysis is based on the former.

In both Fig. 3b and e, $\mu^C$ has a plateau at $\chi_C$ between about 0.25 and 0.35, and the same phenomenon is found at *T* ≤ 5000 K at 50 GPa, and

at even broader temperature range under increasing pressures, up to 8000 K at 600 GPa (see Fig. M4 of the Methods). At 50 GPa, 4000 K (Fig. 3b), $\mu^C$ then decreases rapidly and approaches the ideal behavior at lower $\chi_C$. In contrast, at 400 GPa, 3000 K (Fig. 3e), $\mu^C$ plateaus and reaches a constant value for $\chi_C < 0.12$. The plateaus at low $\chi_C$ were observed at pressures between 200 GPa and 600 GPa and temperatures lower than 3500 K (see Fig. M4 of the Methods). In Fig. 3b, e, the chemical potentials of diamond, $\mu_D$, are indicated by black diamond symbols and horizontal lines. If $\mu^C$ is larger than $\mu_D$ at a given $\chi_C$, diamond formation is thermodynamically favorable.

To rationalize the plateaus, we express the per-atom chemical potential of the C/H mixture as

$$\mu_{mixture}(\chi_C) = \chi_C\mu^C(\chi_C) + (1-\chi_C)\mu^H(\chi_C), \quad (2)$$

and compare it to the ideal solution curve $\mu_{mixture,id} = k_B T(\chi_C\log(\chi_C) + (1-\chi_C)\log(1-\chi_C))$. Figure 3c shows $\mu_{mixture}$ at 50 GPa, 4000 K. Compared with the ideal solution chemical potential (dashed gray curve) which is fully convex, $\mu_{mixture}$ has two edges. One can thus perform a common tangent construction to the $\mu_{mixture}$ curve to find out the coexisting liquid phases. The green line in Fig. 3d indicates the common tangent, and the two green crosses shows the location of the edges. For C/H mixtures with $\chi_C$ between the two atomic ratios ($\chi_C^1 = 0.27$ and $\chi_C^2 = 0.36$ at the condition shown), a liquid-liquid phase separation (PS1) will occur and form two phases with the proportions determined by the lever rule. Here the region between the two edges is not concave but linear, which is because the phase separation has little activation barrier and already occurs during the MD simulations. In other words, a C/H mixture with a carbon fraction that is between the values of $\chi_C^1$ and $\chi_C^2$ will first undergo spontaneous

liquid-liquid phase separation, which explains the corresponding plateaus in $\mu^C$ of Fig. 3b,e.

Furthermore, Fig. 3f shows that, at 400 GPa, 3000 K, $\mu_{mixture}$ at low $\chi_C$ significantly deviates from the ideal solution approximation (dashed gray curve), and one can construct a tangent as plotted in purple. This means that, besides the aforementioned PS1, C/H mixtures at a low C fraction can also phase separate (PS2) into a fluid of mostly hydrogen and another fluid with $\chi_C \approx 0.12$ (purple cross). We show example snapshots of such phase separated configurations collected from the MD simulations in the Supplementary Information. This PS2 explains the plateau of $\mu^C$ at low $\chi_C$ in Fig. 3d, as the carbon concentrations in both phase-separated liquids stay the same, while only the proportions of the two liquids change. Supplementary Movie 1 shows the occurrence of PS2 in MD simulations. This phase separation has immense consequences: at pressures above 200 GPa and temperatures below 3000 K-3500 K, C in C/H mixtures will always have $\mu^C > \mu_D$ even at very low C fraction due to PS2, and the carbon atoms will thus always be under a thermodynamic driving force to form diamond. We refer to these conditions as the "depletion zone".

Figure 3 g presents the thermodynamic phase boundaries, below which diamond formation is possible in C/H mixtures for each indicated carbon atomic fraction. This is obtained by combining the values of $\mu^C(\chi_C)$ in C/H mixtures and $\mu_D$ at a wide range of P-T conditions. For lower and lower $\chi_C$, the boundaries deviate more and more from the $T_m$ of diamond. At P < 100 GPa, the locations of the boundaries are very sensitive to both temperature and pressure, whereas at higher P it is mostly independent of pressure. Figure 3g can also be read in another way: for a certain P-T condition, it gives the minimal carbon ratio required to make diamond formation possible. Notice that the $\chi_C = 0.25$ and $\chi_C = 0.3$ lines almost overlap, which is due to the plateau of $\mu^C$ induced by PS1. The light blue shaded area indicates the depletion zone, where diamond formation is always possible due to PS2. In this zone, carbon atoms will first form a carbon-rich liquid phase, and diamond can nucleate from this phase. Such process is similar with a two-step nucleation mechanism previously revealed in protein systems[38].

Previous experimental measurements are included in Fig. 3g, with the conditions where diamonds were either found (diamond symbols or rectangular regions) or absent (cross symbols) indicated. At lower pressures, our calculations largely agree with the observation of diamond formation for methane in DAC experiments between 2000–3000 K[11] and above 3000 K[12]. We find less agreement with the shock-compression experiments at higher pressures[16,17]. We speculate that the disagreement may be because diamond formation needs to go through the activated nucleation process which may take longer than the short timescale of these rapid compression experiments, or may come from the difficulty in the temperature estimation of these experiments. The hollow diamond symbols in Fig. 3g show the diamond formation conditions from starting materials of more complex compositions: Marshall et al.[15] used epoxy (C:H:Cl:N:O ≈ 27:38:1:1:5) and Kadobayashi et al.[14] used methane hydrate. We find little agreement between Kadobayashi et al.[14] and our phase boundaries, if we compare solely in terms of $\chi_C$ including all atomic species, although the agreement is improved if based on the $\chi_C$ of methane alone, and indeed $CH_4$ may be an intermediate product in the experiment[14]. The liquid-liquid phase separations of C/H mixtures have not been previously observed, but they may be detected from speed of sound, mixed optical spectra, inhomogeneity in the diamond formation reaction, and hydrodynamic instability during compression experiments.

## Discussion

We first computed the melting line of diamond in pure liquid carbon. We then moved on to the C/H mixtures, and showed that they behave like liquids at $T \geq 2500$ K. We finally precisely computed the thermodynamic boundary of diamond formation for different atomic ratios. Notably, we revealed the occurrence of phase

separations in C/H mixtures, which can greatly enhance diamond formation. For PS1, the C/H mixture will phase separate into two liquids with $\chi_C$ of about 0.25 and 0.35. Both liquids have the same $\mu^C$, but their interfacial free energies with the diamond phase are different. Diamond will thus prefer to nucleate from the liquid with the lower interfacial free energy. At 200 < P < 600 GPa and T below 3000 K-3500 K, there is a depletion zone where C/H mixtures at a low C fraction can phase separate (PS2) into a fluid of mostly hydrogen and a more carbon-rich fluid with $\chi_C \approx 0.12$. In this zone, there is always thermodynamic driving force to form diamond from the carbon-rich phase.

Our phase boundaries in Fig. 3g put largely scattered experimental measurements[11,12,15–17] into context, and provide a mechanistic understanding of the thermodynamics involved. They also help gauge the accuracy of the experimental determination of diamond formation conditions, extrapolate between different experiments, and guide future efforts to validate these boundaries.

Note that our boundaries are solely based on the thermodynamic criterion, but the kinetic nucleation rate may play a role particularly in shock-compression experiments. An amount of undercooling may be needed for diamond to nucleate from C/H mixtures within finite time, depending on the magnitude of the nucleation activation barrier. In homogeneous nucleation, the magnitude of the interfacial free energy contribution is crucial[25,32]. In experiments, DAC are in close contact with the fluid samples, so heterogeneous nucleation may happen, which requires less undercooling compared with the homogeneous case. In addition, other elements (e.g. He, N, O) are also prevalent in icy planets, and we suggest future experiments to probe how they affect the phase boundaries of diamond formation.

The "depletion zone" can help explain the difference in the luminosity between Uranus and Neptune. Being similar in size and composition, Neptune has a strong internal heat source but Uranus does not[39]. The "diamonds in the sky" hypothesis[8,11] relates the heat source with diamond formation, but does not explain the dichotomy between the two planets. By comparing the P-T conditions at different depths of the two ice giants from Ref. [40] with our calculated phase boundaries (in Fig. 3g), one can see that a relatively small difference in the planetary profile can drastically change the possibility of diamond formation: At the P-T conditions in Uranus, diamond formation requires about 15% of carbon, which seems unlikely as less than 10% of carbon is believed to be present in its mantle[4]. As such, diamond formation in Uranus may be absent. In contrast, Neptune is a bit cooler so it is much more likely that its planetary profile may have an overlap with the depletion zone; at these conditions C/H mixtures will phase separate (PS2), and diamond formation is thermodynamically favorable regardless of the actual carbon fraction. If there is indeed an overlap, diamonds can in principle form in the depletion zone in the mantle of Neptune, and then sink towards the core while releasing heat. Although the mantle will become increasingly carbon-deprived, the diamond formation in the depletion zone can proceed until all carbon is exhausted. Moreover, the "diamond rain" will naturally induce a compositional gradient inside the planet, which is an important aspect in explaining the evolution of giant planets[41,42].

Our carbon-ratio-dependent diamond formation phase boundaries can help estimate the prevalence and the existence criteria of extraterrestrial diamonds. Neptune-like exoplanets are extremely common according to the database of planets discovered[43], and methane-rich exoplanets are modeled to have a carbon core, a methane envelope, and a hydrogen atmosphere[7]. Our boundaries can put a tight constraint on the structure and composition of these planets. Furthermore, diamond formation and liquid-liquid phase separation play a key role in the cooling process in white dwarfs[10], and thus the precise determination for the onset of phase separation and crystallization is also crucial there.

## Methods

### DFT calculations

DFT is the workhorse of high-pressure equation-of-state calculations and has shown good agreement with several experiments on hydrocarbons and other systems[44–46] for measured thermodynamic properties in particular for Hugoniot curves. Single-point DFT calculations with VASP[47–50] were carried out for configurations with various C/H ratios to generate the training set of the MLP. The simulations were performed with the Perdew-Burke-Ernzerhof (PBE) exchange-correlation functional[51] employing hard pseudopotentials for hydrogen and carbon, a cutoff energy of 1000 eV, and a consistent k-point spacing of 0.2 Å$^{-1}$. In addition, extensive PBE MD simulations for $CH_{16}$, $CH_8$, $CH_4$, $CH_2$, CH, $C_2H$ and $C_4H$ mixtures were performed, and together with previous PBE MD data for methane[52], carbon[30] and hydrogen[53], were used to benchmark the MLP. To approximate the impact of the thermal excitation of the electronic subsystem, we set the electronic temperature equal to the average ionic temperature during the DFT MD calculations as well as in the reference calculations used to train and test the MLP. The convergence tests of DFT and the influence of the electronic temperature are provided in the Supplementary Information.

### Machine learning potential

We generated flexible and dissociable MLPs for the high-pressure C/H system, employing the Behler-Parrinello artificial neural network[54], and using the N2P2 code[55]. The total training set contains 92,185 configurations with a sum of 8,906,582 atoms, and was constructed using a combination of strategies, including DFT MD, random structure searches, adapting previous training sets for pure C[56] and H[53], and active learning. The training set includes a large variety of structures: cubic/hexagonal diamond, graphite, graphane, carbon nanotubes, fullerenes, amorphous carbon, carbon structures with defects, liquid carbon, liquid hydrogen, many hydrogen crystalline polymorphs, hydrocarbon crystals, hydrocarbon liquids with varying carbon concentrations at a wide range of P-T conditions. Details on the construction and the benchmarks of the MLP are provided in the Supplementary Information. Note that the MLP has been extensively benchmarked for high-pressure liquid hydrogen, diamond/liquid carbon, and C/H mixtures based on energetic, thermodynamic and dynamic properties. However, we would like to caution the limitations of the current MLP: The MLP is not applicable to gas-phase hydrocarbons. For low-pressure carbon phases and diamond-graphite transitions, the current MLP has not been extensively tested, and we recommend users to employ the MLP from Ref. [56]. Long-range Van der Waals interactions in liquid methane at low density may be important[57] but are lacking in the current MLP. The comparison between PBE and MLP for structure and dynamic properties such as equation of states, radial distribution functions, diffusivity, vibrational density of states, and bond lifetimes are provided in the Supplementary Information. We recommend checking these comparisons before applying the MLP for a given C/H composition at certain conditions.

### MLP MD simulation details

All MD simulations were performed in LAMMPS[58] with a MLP implementation[59]. The time step was 0.25 fs for C/H mixtures, and 0.4 fs for pure carbon systems.

### Computing the chemical potentials of diamond and pure liquid carbon

We computed $\Delta\mu_D$ using interface pinning simulations[60], which were performed using the PLUMED code[61]. We used solid-liquid systems containing 1,024 C atoms at pressures between 10-800 GPa and at temperatures close to the melting line, employing the MLP. The Nosé-Hoover barostat was used only along the z direction that is perpendicular to the interface in these coexistence

simulations, while the dimensions of the supercell along the x and y directions were commensurate with the equilibrium lattice parameters of the diamond phase at the given conditions. We used the locally-averaged[62] $Q_3$ order parameter[63] for detecting diamond structures, and introduced an umbrella potential to counterbalance the chemical potential difference and constrain the size of diamond in the system. We then used thermodynamic integration along isotherms and isobars[64,65] to extend the $\Delta\mu_D$ to a wide range of pressures and temperatures.

### MLP MD simulation of $CH_4$

The simulation cell contained 7,290 atoms (1,458 $CH_4$ formula units). Each simulation was run for more than 100 ps. The simulations were performed in the NPT ensemble, using the Nosé-Hoover thermostat and isotropic barostat. At each condition, two independent MD simulations were initialized using a starting configuration of either bonded $CH_4$ molecules on a lattice or a liquid. For $T \geq 2500$ K, the two simulations provided consistent statistical properties. These simulations were the basis for the further analysis we performed. For $T < 2500$ K the two runs gave different averages, meaning that under these conditions the system is not ergodic within the simulation time.

### Computing the chemical potentials of C in C/H mixtures

We used two independent methods for computing the chemical potentials of carbon in C/H mixtures at various conditions. The first is the S0 method[37] that uses the static structure factors computed from equilibrium NPT simulations. The S0 method uses the thermodynamic relationship between composition fluctuations and the derivative of chemical potential with respect to concentration, and accounts for both mixing entropy and enthalpy[37]. The simulations were performed on a grid of P-T conditions, P = 10 GPa, 25 GPa, 50 GPa, 100 GPa, 200 GPa, 300 GPa, 400 GPa, 600 GPa, and T = 3500 K, 4000 K, 5000 K, 6000 K, 7000 K, and 8000 K. At each P-T condition, MD simulations were run for systems at varying atomic ratios, on a dense grid of $\chi_C$ from 0.015 to 0.98. The system size varied between 9,728 and 82,944 total number of atoms. We obtained the static structure factors at different wavevectors **k** using the Fourier expansion on the scaled atomic coordinates, i.e.

$$S_{AB}(\mathbf{k}) = \frac{1}{\sqrt{N_A N_B}}\left\langle \sum_{i=1}^{N_A} \exp(i\mathbf{k}\cdot\hat{\mathbf{r}}_{i_A}(t)) \sum_{i=1}^{N_B} \exp(-i\mathbf{k}\cdot\hat{\mathbf{r}}_{i_B}(t)) \right\rangle \quad (3)$$

where AB can be CC, CH and HH, and $\hat{\mathbf{r}}(t) = \mathbf{r}(t)\langle l \rangle_{NPT}/l(t)$ and $l(t)$ is the instantaneous dimension of the supercell. We then determined $S_{CC}^0$, $S_{CH}^0$ and $S_{HH}^0$ by extrapolating $S_{AB}(\mathbf{k})$ to the $\mathbf{k}\to 0$ case using the Ornstein–Zernike form as described in Ref. [37]. Finally, we used numerical integration using Eqn. (3) of the main text to obtain the chemical potential of carbon for different atomic fractions, and get the chemical potential of H using the Gibbs-Duhem equation. All the chemical potential data are presented in Fig. 4.

The second approach is based on the coexistence method, similar to the setup used for computing the chemical potentials of the pure carbon systems. In this case, interface pinning simulations[60,66] were performed on a diamond-C/H liquid coexistence system containing 1024 C atoms and a varying number of H atoms at pressures 0-600 GPa. A snapshot of the coexistence system is provided in the Supplementary Information. The chemical potentials estimated using coexistence are shown in Fig. 4, and the errors shown are the standard errors of the mean estimated from the values of the CV. However, there are other sources of errors that are hard to estimate: finite size effects and ergodicity issues related to the explicit interface; the carbon concentration can vary in the liquid region of the simulation box.

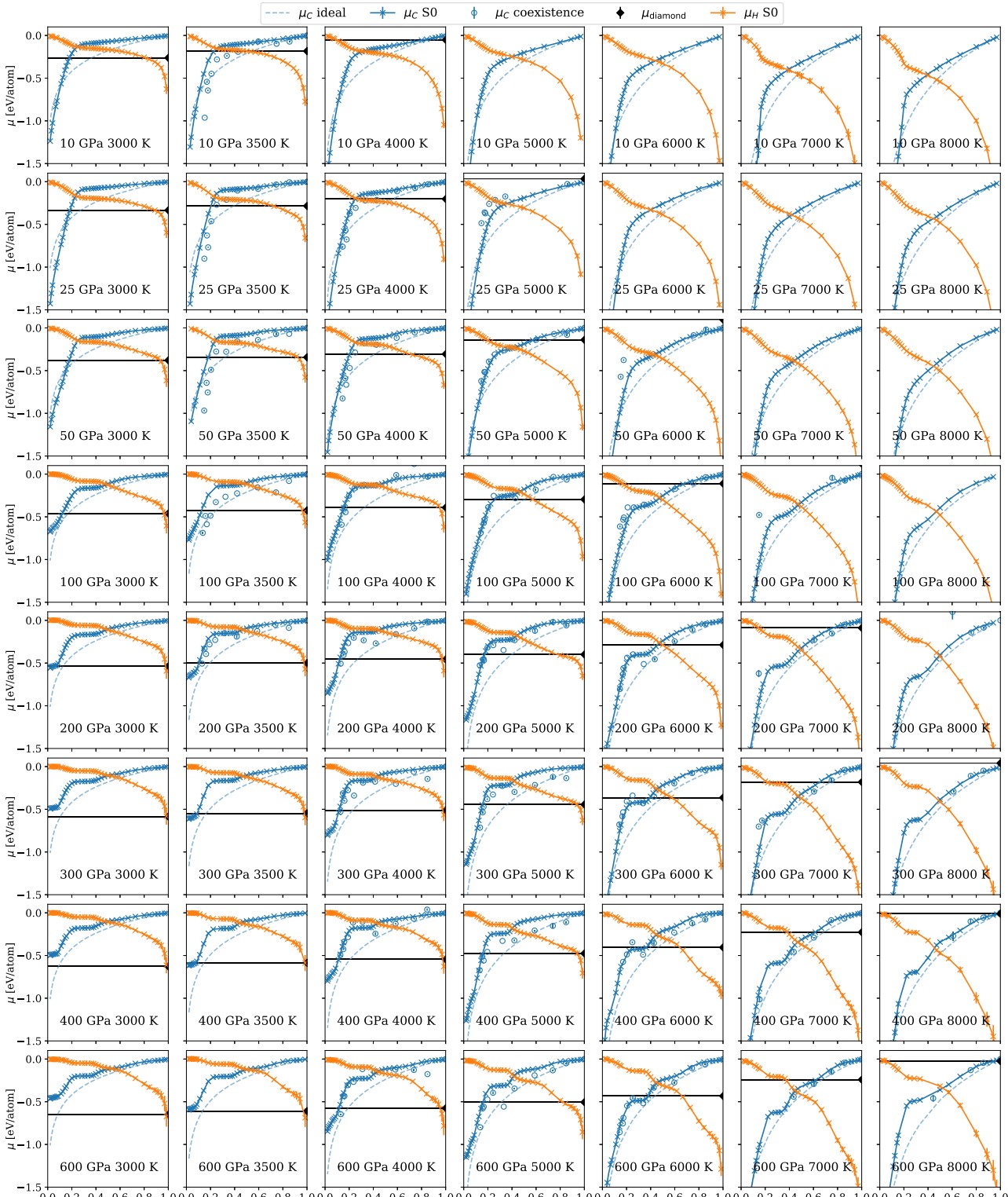

**Fig. 4 | The chemical potentials of carbon and hydrogen in C/H mixtures at different temperature and pressure conditions.** The chemical potentials per carbon atom in C/H mixtures calculated using the S0 method are shown as blue crosses connected by solid curves, the values obtained using the coexistence method are shown as blue hollow symbols connected by dotted curves, and the ideal solution approximation is indicated by blue dashed curves. The chemical potentials per hydrogen atom calculated using the S0 method are shown as orange crosses. The error bars show the standard mean errors. The chemical potential of diamond is indicated by the black diamond symbol and the black horizontal line.

## Data availability

All original data generated for the study, including the MLP, the training set, simulation input files, intermediate data, PYTHON notebook, are in the SI repository https://github.com/BingqingCheng/highp-ch[67].

## Code availability

The MD simulations were performed using the LAMMPS code[58] with a MLP implementation[59].

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

## Acknowledgements
BC thanks Daan Frenkel for stimulating discussions. We thank Aleks Reinhardt, Daan Frenkel, Marius Millot, Federica Coppari, Rhys Bunting, and Chris J. Pickard for critically reading the manuscript and providing useful suggestions. BC acknowledges resources provided by the Cambridge Tier-2 system operated by the University of Cambridge Research Computing Service funded by EPSRC Tier-2 capital grant EP/P020259/1. SH acknowledges support from LDRD 19-ERD-031 and computing support from the Lawrence Livermore National Laboratory (LLNL) Institutional Computing Grand Challenge program. Lawrence Livermore National Laboratory is operated by Lawrence Livermore National Security, LLC, for the U.S. Department of Energy, National Nuclear Security Administration under Contract DE-AC52-07NA27344. MB acknowledges support by the European Horizon 2020 program within the Marie Skłodowska-Curie actions (xICE grant number 894725), funding from the NOMIS foundationand computational resources at the North-German Supercomputing Alliance (HLRN) facilities.

## Author contributions
B.C., S.H., and M.B conceived the idea of studying high-pressure methane; B.C. designed the research; B.C. performed the simulations related to the MLP; B.C., S.H., and M.B. performed the DFT calculations; B.C., S.H., and M.B wrote the paper.

## Competing interests
All authors declare no competing interests.

## Additional information

**Peer review information** : *Nature Communications* thanks the anonymous reviewer(s) for their contribution to the peer review of this work. Peer reviewer reports are available.

