## [Peer Review File · Nature Communications]

Thermodynamics of diamond formation from hydrocarbon mixtures in planetsREVIEWER COMMENTS

Reviewer #1 (Remarks to the Author):

The manuscript by Chen et al. examines the carbon-hydrogen phase diagram at extreme pressure-temperature conditions with the focus on diamond nucleation and growth from carbon-hydrogen mixtures. The authors go beyond using standard first-principles computational methods, which remain limited to relatively small system sizes and potentially failing to reveal real chemical/physical behavior. Among the important and novel findings of this study is the pressure-temperature space where any carbon-hydrogen fluid is thermodynamically driven to crystallize diamond (termed 'depletion zone'). The finding of 'depletion zone' potentially resolves the long-standing question about the apparently different luminosity of Uranus and Neptune, albeit these two planets being largely similar in many other physical parameters.

Overall, the manuscript is clearly written and understandable to a reader without comprehensive background in first-principles computations (such as myself). I recommend this work for publication in Nature Communications with the caveat that its technical details are approved by another reviewer in possession of computational expertise. Below I provide my questions and comments that together amount to a minor revision.

--- Is it possible to make a quantitative estimate of the diamond production rate in the interiors of Uranus and Neptune in or above the 'depletion zone' given their expected carbon contents? If yes, can this rate be extrapolated in the past and compared to the overall carbon budget of these planets.

--- 'In our chemical bond analysis, a C-C bond is identified whenever the distance between a pair of carbon atoms is within 1.6 Å, and a C-H bond is defined using a cutoff of 1.14 Å. These cutoffs correspond to the first neighbor shells of the C-C and C-H radial distribution functions, respectively.' A typical C-C bond in hydrocarbons is <1.6 Angstrom at ambient conditions and thus must be smaller at high pressure. Could you please clarify the choice of the 1.6 Å cutoff (instead of, say, a variable or a smaller value). Please, also comment on the possible presence of carbon-carbon double and triple bonds in the simulations and how these affect the findings of the present work.

--- Fig. 1 (c) – x-axis (n_s) needs a scale.

--- 'DFT studies coupled with harmonic approximations have predicted a variety of hydrocarbon crystals to be stable at $P \leq 300$ GPa [18–22], but these studies are restricted to low temperatures as the melting lines of hydrogen and methane are below 1000 K and 2000 K, respectively, while harmonic approximations break down completely for these liquids' In my opinion, references are in order for the melting curves of hydrogen and methane.

Reviewer #2 (Remarks to the Author):

The manuscript by Cheng and co-authors presents a thorough computational study, which combines thermodynamics and kinetics, of diamond formation from pure carbon and hydrocarbons, up to 600 GPa and 8000K.

The study is very thorough, leveraging on methods that are standard in both computational statistical mechanics and machine-learning potential-energy-surface (PES) fitting. The only non-standard method is the recently introduced (by the first author of this manuscript) "S0" method. The merit of the manuscript is to integrate the study of pure-carbon with hydrocarbons in order to offer an updated answer to the long-standing "diamond in the sky?" question. The crucial step forward is to go beyond the ideal C,H mixture model assumed in the past.

If the atomistic interaction model used in this work is reliable, then the results are credible and important.

My criticism, below, is therefore focused on both the ab initio modeling choice and the MLP trained on top o it.

1 - How much realistic is Born-Oppenheimer MD via DFT for liquid C at the temperatures and pressures here studied? Liquid C must be metallic at those conditions, and the high density suggests also that an approximate treatment of core-electrons such as PAW (as opposed to an all-electron, full potential treatment) might be problematic. This is a crucial validation point as the comparison between the here presented results with experiments (in particular Kraus and Hartley) does not look consistent. I am ready to trust more the theory compared to very difficult experiments, IF the theory's scope is challenged and subsequently validated. Since there is no methodological advance for this manuscript, then everything is played at the level of predictions' reliability.

2 - How reliable is the MLP itself, compared to the reference DFT? I see, as it is typical for MLPs, a (very thorough, in particular in the impressive SI) statistical validation of the MLP, plus a report on random structure search. This is nice, but phase coexistence and nucleation kinetics depend on the accurate description of the subtle chemistry of C, with its bonding flexibility. What about formation enthalpy and vibrational properties (spectra) of small hydrocarbon molecules, including aromatic ones? What about the diamond to graphite transition barrier and the torsional energy of single, aromatic, double bonds? The latter point was for instance crucial for LCBOP when discussing the liquid-liquid phase transition. Obviously, such validation is more compelling if such molecules and structures are not explicit part of the training set.

3 - Why a newly trained MLP is necessary at all, considering the extensive work presented by Rowe et al. (ref. 56)? I.e., why does one need Behler-Parrinello symmetry functions and the related NN machinery vs the SOAP-GAP solution? I would like to see some discussion of this. It is nice that, compared to the unsystematic times of empirical potentials when designing a training set and a training procedure was an art (quoting a famous paper by Donald Brenner), MLPs can be systematically trained by (in principle) everybody. However, I think one should justify why one more MLP needs to be created.

4 - In this respect, is there any plan to provide to the future readers the full training set(s) and procedure (in terms of actual data and scripts) in a localized repository (matbench, NOMAD, etc.)? We are in the era of FAIR data: users need to be able to reproduce results without barriers in order to accelerate the development of models and understanding.

5 - I would like to understand the implications of Fig2b better. The 1-bond region suggests that on average one has C-C dimers, probably duly saturated by hydrogens. I suspect that C_2H_x is not the most probable species, though, but rather one observes a mixture of CH_4 and short chains. What is the distribution of C-chain length in the 1-bond region (e.g., at few selected state points)? Crucially, how to explain/characterize the narrow strip of 2-bonds liquid, separating the 1-bond and 3-bond liquids? Is it a phase transition? A first order one? Such boundary region seems even to end in a critical point around 500 GPa.

6 - I do not recognize the wording "fully dissociable" attribute to the MLP presented in the manuscript. I understand that there is a whole class of non-dissociable (normally called "non reactive", I believe, i.e.,) potentials where bonds do not break by construction, widely used in soft matter and biophysics. However, I am not keen on recognizing degrees of dissociability, such that the present potential is fully dissociable and others (e.g.?) only partially. So, my suggestion is that it is fine to point out that this MLP treats bond breaking and forming, but I would leave out the unclear "fully", unless I am convinced otherwise.

I will be happy to reconsider the publication of this manuscript after the above points (in particular 1, 2, and 5).

Reviewer #3 (Remarks to the Author):

Cheng et. al. studied the diamond formation out of pure liquid carbon and the P-T phase boundaries using machine learning force field. I do not think that the manuscript reaches the publication level required by Nature Communications. Below are my comments:

1. The title is misleading. When talking about diamond formation from hydrocarbons, we would like to know both the kinetics and thermodynamics, e.g., how diamonds are generated out of hydrocarbons

at the atomistic scale. But this work only provides P-T conditions for the possible existence of diamonds, so the work doesn't match my excitement and expectation when reading the title. Furthermore, many experiments studied C/H mixtures, and gave inconsistent P-T conditions for the diamond formation, as shown in the introduction. This work agrees with some experiments but not with others, which the authors attributed to "the kinetic effects". However, the authors did not give any evidence about kinetics to support this claim.

2. For the diamond nucleation out of pure liquid carbon, the authors used metadynamics to obtain the free energy barrier for the diamond formation. However, the collective variable (CV) is simply the number of atoms that have the diamond structure, which may not be a good assumption if the carbon nucleation has more than one step. In fact, the authors also mention the two-step nucleation mechanism on page 5, ref. 42. The possible intermediate structures may bring some problems when defining CVs to calculate energy barriers. Moreover, the intermediate structures can be related to non-classical nucleation mechanisms, which means that eq. 1 from the classical nucleation theory is problematic. Furthermore, the surface energy is "extrapolated to other conditions using a linear fit in both P and T ." Is it reasonable? With so many assumptions and approximations, the conclusions here are uncertain.

3. In Figs. S4 and S10, it seems that the radial distribution functions (RDFs) obtained from MLP do not match well with the DFT data at some conditions. Anything wrong with MLP? Furthermore, besides energies and RDFs, can the authors also validate the dynamics of C-C and C-H bonds, e.g., the bond lifetime, which are important for hydrocarbon/diamond reactions?

4. What are the statistical errors of the chemical potentials calculated here, considering the statistical features of machine learning? I suspect that the "liquid-liquid phase separation" may not exist because of the statistical uncertainty.

5. Does μ_{mixture} (free energy as in Fig. 3 c,f) include the mixing entropy in the calculations? How?

6. The coexistence method was used to calculate the chemical potentials, and the authors said it agrees with the S0 method. However, I can see some differences as shown in Fig. 3. Considering this, is the linear fitting for the regions of liquid-liquid phase separation too arbitrary?

7. What is the highest nucleation rate in Fig. 1d? Is it possible to directly simulate the nucleation event at the highest nucleation rate? Or some enhanced method without predefined collective variables?

8. How does the production rate of diamonds change with the carbon ratio? It would be very interesting to find the maximum production rate.

9. How would the liquid-liquid phase separation of C/H mixtures affect the diamond formation?

10. "We show example snapshots of such phase separated configurations collected from the MD simulations in the Supplementary Information." It is better to show some videos for the phase separation.

11. No unit in the legend of Figure 1c.

We thank the referees for their careful reading of our manuscript. We have made a number of changes to the manuscript in response to their comments, and have highlighted these in blue in the revised version of the text. In what follows, we respond to each of the points raised by the referees.

Reviewer #1 (Remarks to the Author):

The manuscript by Chen et al. examines the carbon-hydrogen phase diagram at extreme pressure-temperature conditions with the focus on diamond nucleation and growth from carbon-hydrogen mixtures. The authors go beyond using standard first-principles computational methods, which remain limited to relatively small system sizes and potentially failing to reveal real chemical/physical behavior. Among the important and novel findings of this study is the pressure-temperature space where any carbon-hydrogen fluid is thermodynamically driven to crystallize diamond (termed 'depletion zone'). The finding of 'depletion zone' potentially resolves the long-standing question about the apparently different luminosity of Uranus and Neptune, albeit these two planets being largely similar in many other physical parameters.

Overall, the manuscript is clearly written and understandable to a reader without comprehensive background in first-principles computations (such as myself). I recommend this work for publication in Nature Communications with the caveat that its technical details are approved by another reviewer in possession of computational expertise. Below I provide my questions and comments that together amount to a minor revision.

AUTHORS:

We thank the Referee for examining our work and giving a positive assessment.

--- Is it possible to make a quantitative estimate of the diamond production rate in the interiors of Uranus and Neptune in or above the 'depletion zone' given their expected carbon contents? If yes, can this rate be extrapolated in the past and compared to the overall carbon budget of these planets.

AUTHORS:

In principle it is possible to quantitatively estimate diamond formation in the planetary interiors based on our computed thermodynamic phase diagram (Fig.3g). In practice, this requires an accurate planetary interior model that contains pressure, temperature and hydrogen/carbon concentrations at different depths of the planets. Such a model should account for compositional gradients at different P,T conditions. Such a model is not currently available, partly because of the uncertainty in the observations/measurements of the planets, and partly because certain physical processes such as possible liquid-liquid phase separations and diamond formation are missing from the state-of-the-art planetary models. Extrapolating to the past is even more difficult because there is more uncertainty about the evolution of the planets.

--- 'In our chemical bond analysis, a C-C bond is identified whenever the distance between a pair of carbon atoms is within 1.6 Å, and a C-H bond is defined using a cutoff of 1.14 Å. These

cutoffs correspond to the first neighbor shells of the C-C and C-H radial distribution functions, respectively.'

A typical C-C bond in hydrocarbons is <1.6 Angstrom at ambient conditions and thus must be smaller at high pressure. Could you please clarify the choice of the 1.6 A cutoff (instead of, say, a variable or a smaller value). Please, also comment on the possible presence of carbon-carbon double and triple bonds in the simulations and how these affect the findings of the present work.

AUTHORS:

The 1.6 A cutoff here is selected based on the first neighbor shell of the C-C radial distribution functions (RDFs), i.e. the first minima in the RDFs. As shown in the RDFs of CH₄ below, this is a feature that is quite consistent across a wide temperature-pressure range. Moreover, as the Referee correctly pointed out, a typical C-C bond in hydrocarbons is less than 1.6 Angstrom, so having a cutoff that is 1.6 A ensures that thermal fluctuations of a C-C bond will not cause the bond to be misidentified as broken.

Due to the high energy of the systems at the high-pressure high-temperature conditions, the unsaturated carbon bonds are unlikely to exist.

We have added these explanations in the main text and the SI (Sec. IV D) of the paper.

--- Fig. 1 (c) – x-axis (n_s) needs a scale.

AUTHORS:

We have now added this. n_s is the number of atoms in the solid nucleus.

--- 'DFT studies coupled with harmonic approximations have predicted a variety of hydrocarbon crystals to be stable at $P \leq 300$ GPa [18–22], but these studies are restricted to low temperatures as the melting lines of hydrogen and methane are below 1000 K and 2000 K, respectively, while harmonic approximations break down completely for these liquids' In my opinion, references are in order for the melting curves of hydrogen and methane.

AUTHORS:

We have now added the references for the melting curves of hydrogen and methane.

Reviewer #2 (Remarks to the Author):

The manuscript by Cheng and co-authors presents a thorough computational study, which combines thermodynamics and kinetics, of diamond formation from pure carbon and hydrocarbons, up to 600 GPa and 8000K.

The study is very thorough, leveraging on methods that are standard in both computational statistical mechanics and machine-learning potential-energy-surface (PES) fitting. The only non-standard method is the recently introduced (by the first author of this manuscript) "S0" method. The merit of the manuscript is to integrate the study of pure-carbon with hydrocarbons in order to offer an updated answer to the long-standing "diamond in the sky?" question. The crucial step forward is to go beyond the ideal C,H mixture model assumed in the past. If the atomistic interaction model used in this work is reliable, then the results are credible and important.

My criticism, below, is therefore focused on both the ab initio modeling choice and the MLP trained on top o it.

AUTHORS:

We thank the Referee for carefully examining our work. We have added additional validations of the MLP in the revised version of the paper.

1 - How much realistic is Born-Oppenhemier MD via DFT for liquid C at the temperatures and pressures here studied? Liquid C must be metallic at those conditions, and the high density suggests also that an approximate treatment of core-electrons such as PAW (as opposed to an all-electron, full potential treatment) might be problematic. This is a crucial validation point as the comparison between the here presented results with experiments (in particular Kraus and Hartley) does not look consistent. I am ready to trust more the theory compared to very difficult experiments, IF the theory's scope is challenged and subsequently validated. Since there is no methodological advance for this manuscript, then everything is played at the level of predictions' reliability.

AUTHORS:

The Referee asked about the reliability of a 4 electron PAW potential for carbon at high pressure. We respond thusly:

Our DFT calculations are performed with the “hard” version of the PAW potentials. The accuracy of the carbon PAW potentials at high pressure has been the object of a study by Benedict et al. [Benedict et al. PRB 224109 (2014)], these authors reported that the 4 electron hard PAW from the VASP library compared well with a 6 electron (all-electron) PAW up to densities of 60 g/cc, corresponding to carbon-carbon distance of 1.3 atomic units (0.7 Angstroms). This is well beyond the range of densities encountered in this study.

To clarify this point we added this paragraph to the SI in section I:

“These consist of the hard version of the PAW potentials. The accuracy of this carbon PAW potential at high pressure has been the object of a study by Benedict et al. [Benedict et al. PRB 224109 (2014)], these authors reported that the 4 electron hard PAW from the VASP library compared well with a 6 electron (all-electron) PAW potential up to densities of 60 g/cc, corresponding to carbon-carbon distance of 1.3 atomic units (0.7 Angstroms).”

The referee also asked about the realtiness of the Born-Oppenheimer approximation in metallic carbon. To this we respond:

The ground truth in this study is density functional theory (DFT) within the PBE approximation to the exchange-correlation potential and within the Mermin finite-temperature formalism. Born-Oppenheimer DFT Molecular Dynamics is the workhorse of high pressure equation-of-state calculations and has shown good agreement with several experiments on hydrocarbons and other systems [Knudson et al. JAP **129**, 210904 (2021), Millot et al. Nature Physics **14**, 297 (2018), Kim et al. PRL **129**, 015701 (2022)] for measured thermodynamic properties in particular for Hugoniot curves in pressure-density space and in pressure-temperature space when the temperature is inferred with pyrometry.

Regarding the comparison between our computed melting curve with Eggert et al. [Nature Physics **6**, 40 (2010)], the agreement is good (capturing the reentrant behavior of the diamond melting line at high pressure). The experiments have uncertainties and now we have plotted the uncertainties in Fig.1.a. There is an offset in temperature between the BOMD calculations of the melting line and the experimental melting line and one possible explanation is systematic uncertainties in their temperature estimates. A recent set of experiments (as yet unpublished) that are in closer agreement with our DFT BOMD melting line supports this explanation [Millot, private communication].

For the diamond formation in C/H mixtures, experiments are scattered and contradicting with each other. There, possible reasons for the disagreement are kinetics effects and temperature inference for the reported thermodynamic states (PT points). In particular, the temperatures reported by Hartley et al. were inferred from an equation of state (SESAME 7171) that was used outside the range of pressure and temperature where it was validated ($P < 50$ GPa) [Knudson et

al. JAP **129**, 210904 (2021)] possibly leading to large systematic uncertainties in their temperature inference.

We added this paragraph to the SI in section II:

“Born-Oppenheimer DFT Molecular Dynamics is the workhorse of high pressure equation-of-state calculations and has shown good agreement with several experiments on hydrocarbons and other systems [Knudson et al. JAP **129**, 210904 (2021), Millot et al. Nature Physics **14**, 297 (2018), Kim et al. PRL **129**, 015701 (2022)] for measured thermodynamic properties in particular for Hugoniot curves in pressure-density space and in pressure-temperature space when the temperature is estimated with pyrometry.”

2 - How reliable is the MLP itself, compared to the reference DFT? I see, as it is typical for MLPs, a (very thorough, in particular in the impressive SI) statistical validation of the MLP, plus a report on random structure search. This is nice, but phase coexistence and nucleation kinetics depend on the accurate description of the subtle chemistry of C, with its bonding flexibility. What about formation enthalpy and vibrational properties (spectra) of small hydrocarbon molecules, including aromatic ones? What about the diamond to graphite transition barrier and the torsional energy of single, aromatic, double bonds? The latter point was for instance crucial for LCBOF when discussing the liquid-liquid phase transition. Obviously, such validation is more compelling if such molecules and structures are not explicit part of the training set.

AUTHORS:

The Referee asked about the reliability of our MLP, particularly regarding the subtle chemistry of C. We respond as the following:

- The training set of our MLP explicitly includes a large variety of carbon structures/molecules with single, aromatic, double bonds: cubic/hexagonal diamond, graphite, graphane, carbon nanotubes, fullerenes, amorphous carbon, and defected structures. Indeed, all of the training structures of the GAP-20 model for carbon (Ref. 57, <https://aip.scitation.org/doi/pdf/10.1063/5.0005084>) are included in the training of our MLP. The GAP-20 carbon MLP is regarded to well-capture the chemistry of carbon, because it has been thoroughly benchmarked using metrics such as formation energies, radial distribution functions and the torsional energies of carbon bonds. Therefore, the training set is comprehensive enough to capture the chemistry of C.
- Although the MLP is able to describe the enthalpies of diamond and graphite, we do not study the diamond-graphite transition in the present paper, and we do not discuss the liquid-liquid phase transition in liquid carbon. As such, these topics, while extremely interesting, are out-of-scope for the present paper.
- Our MLP is for bulk C/H mixtures at $10 \text{ GPa} \leq P \leq 600 \text{ GPa}$ and $2500 \text{ K} \leq T \leq 8000 \text{ K}$, and is not applicable for small hydrocarbon molecules in the gas phase. The gas-phase molecules are also not relevant for our study.
- In the SI, we have already included many validations of the MLP, based on equations of states, radial distribution functions, and diffusivities of various C/H compositions.
- Nevertheless, we have added new validations based on vibrational density of states (vDOS). vDOS is a particularly stringent test of the potential energy surface, and the

MLP has good agreement with PBE across a wide range of relevant conditions and for different C/H compositions.

To clarify these points, we have expanded the explanation of the training set in the Methods section and in the SI (Sec III A), and add vDOS to SI (Sec. IV D,E,F).

3 - Why a newly trained MLP is necessary at all, considering the extensive work presented by Rowe et al. (ref. 56)? I.e., why does one need Behler-Parrinello symmetry functions and the related NN machinery vs the SOAP-GAP solution? I would like to see some discussion of this. It is nice that, compared to the unsystematic times of empirical potentials when designing a training set and a training procedure was an art (quoting a famous paper by Donald Brenner), MLPs can be systematically trained by (in principle) everybody. However, I think one should justify why one more MLP needs to be created.

AUTHORS:

The new MLP that we trained for this study is absolutely necessary because (to the best of our knowledge) all the previous MLPs cannot describe the C/H mixture with variable compositions at high-pressures (10-600 GPa). The ones by Rowe et al. (GAP-20) is for pure carbon only, and only works for pressures less than about 50 GPa (liquid carbon of density 3.5 g/mL). We have highlighted these justifications in the manuscript.

4 - In this respect, is there any plan to provide to the future readers the full training set(s) and procedure (in terms of actual data and scripts) in a localized repository (matbench, NOMAD, etc.)? We are in the era of FAIR data: users need to be able to reproduce results without barriers in order to accelerate the development of models and understanding.

AUTHORS:

We agree completely. We have already uploaded all the data needed to reproduce the results, including training set, trained MLP and the input files to train the MLP in the GitHub repository: <https://github.com/BingqingCheng/highp-ch>

BingqingCheng / highp-ch Private

<> Code Issues Pull requests Actions Projects Security Insights Settings

main 1 branch 0 tags Go to file Add file Code

BingqingCheng forgot to add plumed.dat 4b31ab6 on 2 Aug 6 commits

gdr	added gdr	2 months ago
interface-pinning-diamond-liquidca...	add files	3 months ago
metad-diamond-in-liquidcarbon	forgot to add plumed.dat	2 months ago
mlp-md-example	add files	3 months ago
nnp	add new train set for V2	2 months ago
npt-CxHy	add files	3 months ago
scripts	add files	3 months ago
train-set	add new train set for V2	2 months ago
vasp-input	add vasp input	2 months ago
README.md	Initial commit	3 months ago

For now the repository is private, and we will make it public upon the acceptance of the manuscript.

In the past, we have always followed this practice, and made available our MLPs (with training sets) for water, superionic water, high-pressure hydrogen.

5 - I would like to understand the implications of Fig2b better. The 1-bond region suggests that on average one has C-C dimers, probably duly saturated by hydrogens. I suspect that C₂H_x is not the most probable species, though, but rather one observes a mixture of CH₄ and short chains. What is the distribution of C-chain length in the 1-bond region (e.g., at few selected state points)? Crucially, how to explain/characterize the narrow strip of 2-bonds liquid, separating the 1-bond and 3-bond liquids? Is it a phase transition? A first order one? Such boundary region seems even to end in a critical point around 500 GPa.

AUTHORS:

The Referee asked for more clarifications for the implications of Fig2b. In general, the C-C bond lifetime is short (less than about 1 ps), across the whole conditions of Fig.2(b-e), 10 GPa ≤ P ≤ 600 GPa and 2500 K ≤ T ≤ 7000 K. This means that hydrocarbon chains in the systems decompose and form quickly. The C/H mixture is better characterized as a liquid made of C and H atoms, rather than a polymer mixture. To reveal this further, in the figure below we show the distribution for the number of C-C bonds that each carbon atom has. It can be seen that, at all conditions considered, the distribution is rather broad than sharply peaked at a specific bond number. In other words, if on average each carbon atom has 1 bond, this means some C atoms have one bond, some have two bonds, some have zero bond, and so on. All these “bonds” are extremely transient.

The Referee also asked about the narrow strip of 2-bond liquid. Indeed, the original choice of the color scheme (matplotlib, cmap='terrain') in the Fig.2.b made the 2-bond region look particularly narrow because of the drastic color gradient at $n=2$. Now we have changed the color scheme (cmap='rainbow') to eliminate such effects.

Now it can be seen that the average number of C-C bonds varies gradually at the conditions considered here: the number of bonds are higher at higher pressures as well as lower temperatures. There is no first-order phase transition for this CH₄ system, judging from the average number of C-C bonds.

The 2-bond region gets narrower at low temperatures. This can be rationalized using the results from Fig.3(a,d): the CH₂ composition ($X_c=0.33$) is not thermodynamically favorable ($d\mu/dx_c \sim 0$), and it tends to undergo a liquid-liquid phase separation at lower temperatures (the PS1 mechanism). However, the analysis based on the chemical potentials of the C/H mixture presented in Fig.3 is much more rigorous and quantitative than the observation about the 2-bond region, so we refrain from drawing any direct conclusions from the latter.

We have expanded the SI (Sec V H) and the main text to incorporate the discussion above.

6 - I do not recognize the wording "fully dissociable" attribute to the MLP presented in the manuscript. I understand that there is a whole class of non-dissociable (normally called "non reactive", I believe, i.e.,) potentials where bonds do not break by construction, widely used in soft matter and biophysics. However, I am not keen on recognizing degrees of dissociability, such that the present potential is fully dissociable and others (e.g.?) only partially. So, my suggestion is that it is fine to point out that this MLP treats bond breaking and forming, but I would leave out the unclear "fully", unless I am convinced otherwise.

AUTHORS:

We have removed the word "fully".

I will be happy to reconsider the publication of this manuscript after the above points (in particular 1, 2, and 5).

Reviewer #3 (Remarks to the Author):

Cheng et. al. studied the diamond formation out of pure liquid carbon and the P-T phase boundaries using machine learning force field. I do not think that the manuscript reaches the publication level required by Nature Communications. Below are my comments:

AUTHORS:

We thank the Referee for taking the time to critically examine our work and provide constructive suggestions. We have revised the manuscript following the comments of the Referee.

1. The title is misleading. When talking about diamond formation from hydrocarbons, we would like to know both the kinetics and thermodynamics, e.g., how diamonds are generated out of hydrocarbons at the atomistic scale. But this work only provides P-T conditions for the possible existence of diamonds, so the work doesn't match my excitement and expectation when reading the title. Furthermore, many experiments studied C/H mixtures, and gave inconsistent P-T conditions for the diamond formation, as shown in the introduction. This work agrees with some experiments but not with others, which the authors attributed to "the kinetic effects". However, the authors did not give any evidence about kinetics to support this claim.

AUTHORS:

We are confused by this comment from the Referee. Indeed we studied both the thermodynamics and kinetics of diamond formation:

- Thermodynamics: Our motivation to study diamond formation from high-pressure hydrocarbons is to probe the existence of such processes in planets. Crucially, planets have billions of years to evolve and the C/H mixtures inside the planets have a long time to equilibrate, so thermodynamic equilibrium or quasi-thermodynamic equilibrium are good assumptions for the states of hydrocarbons in planets.

To provide the thermodynamic picture of diamond formation from hydrocarbons, we computed the thermodynamic driving force $\Delta\mu$ for the carbon to form the diamond phase from the C/H liquid mixture at a wide range of P-T conditions and with different carbon fractions. In the thermodynamic phase diagram (Fig.3.g), the phase boundary where $\Delta\mu = 0$ provides the P-T conditions for the possible existence of diamonds, taking into account all the thermal effects including the mixing entropy and the enthalpy of the C/H liquid mixture, as well as the thermal fluctuations of the diamond phase. This phase diagram shows under which conditions diamond formation is possible, and under which conditions it is impossible. These conditions can then be compared to the internal conditions of planets.

It is worth mentioning that, to the best of our knowledge, this is the first time that chemical potentials of mixtures are computed at the ab initio-accuracy level.

- Kinetics: We computed the kinetic nucleation rates for diamond formation in pure liquid carbon at a wide range of thermodynamic conditions. Such nucleation rates account for the activation barrier and the kinetic prefactor of nucleation. The nucleation rates can inform us that at a wide range (P,T) conditions whether liquid carbon will solidify into diamond or remain metastable during a given timescale.

In principle, it may also be possible to compute the nucleation rate in the C/H mixture using the same framework, although the simulations will be very expensive considering

the slower kinetic factor and to perform simulations at various C fractions. Alternatively, one can extrapolate the pure carbon case to the C/H mixture, e.g. Ghiringhelli et al. computed the nucleation rates at two state points using the LCBOP model (Ref 25, see Fig.1d), and then extrapolated the rates to other thermodynamic conditions and carbon fractions (based on the ideal solution assumption). In comparison, our computations are much more quantitative because the nucleation rates are based on MLPs with ab initio-accuracy and were computed at many conditions, and our chemical potentials of carbon were computed accurately.

Finally, It is worth noting that, using the machine learning potential that we constructed, it is possible to run dynamic compression simulations. However, the situation in the planets is closer to the thermodynamic equilibrium picture shown in Fig.3.g, than the dynamic compression simulations/experiments that live on the nanosecond timescale.

Regarding the disagreement with experiments, we speculate that the possible cause may be due to the kinetic effects in the rapid compression and/or the difficulty in the temperature estimation of these experiments. Here the kinetic effects refer to that these experiments are fast but the diamond nucleation may take longer time to happen. The temperature estimation of these experiments are difficult and can be inaccurate, because the temperatures are often not directly measured, but rather inferred from a specific equation-of-state model that may or may not be precise in a particular regime of pressure and temperature. In particular, the temperatures reported by Hartley et al. were inferred from an equation of state (SESAME 7171) that was used outside the range of pressure and temperature where it was validated ($P < 50$ GPa) [Knudson et al. JAP **129**] possibly leading to large systematic uncertainties in their temperature inference. Moreover, even the occurrence of diamond formation is nontrivial to verify for the reactions happening inside a diamond anvil cell. Due to all these complications, we can only speculate the cause of the disagreement. We have rewritten the discussion to make it clear the speculative nature of our explanation for the experimental results.

2. For the diamond nucleation out of pure liquid carbon, the authors used metadynamics to obtain the free energy barrier for the diamond formation. However, the collective variable (CV) is simply the number of atoms that have the diamond structure, which may not be a good assumption if the carbon nucleation has more than one step. In fact, the authors also mention the two-step nucleation mechanism on page 5, ref. 42. The possible intermediate structures may bring some problems when defining CVs to calculate energy barriers. Moreover, the intermediate structures can be related to non-classical nucleation mechanisms, which means that eq. 1 from the classical nucleation theory is problematic. Furthermore, the surface energy is "extrapolated to other conditions using a linear fit in both P and T ." Is it reasonable? With so many assumptions and approximations, the conclusions here are uncertain.

AUTHORS:

The Referee suggested that the choice of the CV may not be good if there is a two-step nucleation process. We respond as follows:

- We performed direct MD simulations to probe the nucleation process at deeply undercooled regime, and did not observe any sign of the two-step mechanism (see below).
- In our metadynamics simulations, the free energy profiles were easy to converge with little hysteresis, which suggests that the nucleation process here is a simple one-step process. If there was a two-step transition during the nucleation process, the chosen CV (based on the number of diamond-like atoms) may not capture all the transition states. If that had happened, in metadynamics simulations using this CV, one would see a lot of hysteresis and the free energy profile will be very difficult to converge. One would need another CV to run enhanced sampling simulations in the 2D space to properly converge the free energies. However, this is not the case here.

Regarding the extrapolation of the surface energy to obtain the nucleation barrier, we have the following justifications:

- The most relevant conditions for studying nucleation rates are between about 5% to 20% of undercooling, because at lower undercooling the nucleation rate is negligibly small and at higher undercooling nucleation will happen almost instantly so it is not particularly meaningful to compute the exact rates. For the relevant range of conditions, we have indeed run metadynamics simulations and computed the free energy profiles directly (Fig.S22). These conditions include: 25GPa, 50GPa, 100GPa, 200GPa, 300GPa, 400GPa, 500GPa, 600GPa, and at different undercoolings ranging between 8% and 20%. As such, for the relevant range we are indeed performing interpolation rather than extrapolation within this wide range of conditions. Moreover, in many of these simulations, the nucleation barriers were sampled directly.
- Interfacial free energies often have a linear dependence in temperature (e.g. Cahn Hilliard model <https://aip.scitation.org/doi/pdf/10.1063/1.1744102>). The pressure dependence is less clear but for the case of diamond-liquid carbon interface the pressure dependence was found to be rather weak anyways. So the linear fit in both P and T is reasonable. Moreover, Ghiringhelli et al. (Ref 25) extracted the surface energy at two state points (A=(30GPa, 3750K) and B=(85GPa, 5000K)), and then linearly extrapolated the surface energy to other thermodynamic conditions using the fraction of threefold coordinated carbon atoms. We also tried using this fraction in the linear fit, but did not find it to improve the quality of the fit, so we finally used the linear fit based on P and T .

We have expanded the manuscript and the SI to include the discussions above.

3. In Figs. S4 and S10, it seems that the radial distribution functions (RDFs) obtained from MLP do not match well with the DFT data at some conditions. Anything wrong with MLP? Furthermore, besides energies and RDFs, can the authors also validate the dynamics of C-C and C-H bonds, e.g., the bond lifetime, which are important for hydrocarbon/diamond reactions?

AUTHORS:

Figs. S4 compares the carbon-carbon radial distribution function of pure liquid carbon computed from NVT simulations using DFT and MLP. The discrepancy can be seen at 6000K, e.g. at $\rho=5.7, 6.1,$

6.5g/mL. This is because the liquid carbon at 6000K and density close to 6g/mL tends to solidify during the NVT simulations, and the affected conditions can be seen from the points of abnormally low PE in the figure below. This solidification happens at different timesteps during the NVT simulations, causing the MLP and DFT RDFs to look different.

Fig. S10 shows the C-C RDFs for CH₄. The initial configuration of the DFT simulations is methane molecules on a bcc lattice. In hindsight, this was not a good choice because the system can get stuck in this local minima during very short MD simulations: The bcc methane is not the stable phase and should become liquid at $T \geq 2000$ K (above the melting line). At $T \geq 4000$ K the bcc lattice always melts, but at $T = 3000$ K the system is not ergodic. At $T = 3000$ K, the difference between the MLP and the DFT RDFs for CH₄ is thus again due to the ergodicity problem and the different onset of phase transition. For example, at $T = 3000$ K, $\rho = 1.5$ g/mL, the final configuration is still solid.

Snapshots from DFT MD simulation of CH₄ at 3000K, $\rho = 1.5$ g/mL

The RDFs also differ significantly at 4.50 g/mL, but this corresponds to pressures higher than 1000GPa, which we do not consider in the present study.

For the other C/H compositions including CH₁₆, CH₈, CH₂, CH, C₂H, C₄H, the initial configurations were chosen to be a randomly generated structure, and such ergodicity problems did not appear.

To summarize, at relevant conditions, the difference in the RDFs in a few conditions is due to the ergodicity problem of short MD simulations. It is not related to the accuracy of the MLP. The reasons for the discrepancy was explained in the (very long) SI. However, we understand that many readers will not go through all the details in the SI, and the previous figures on the RDFs may be misleading. As such, in the revised SI, we kept the discussion of the discrepancy in the text, but removed the RDFs for liquid carbon at 6000K and the RDFs for CH₄ at 3000K as these results are not physically meaningful. We also removed the results for CH₄ for $\rho \geq 4.0 \text{ g/mL}$ (which corresponds to $P \geq 800 \text{ GPa}$), as such pressure conditions are not relevant for our study. We hope in this way the SI is more informative and a bit simpler.

Moreover, we have added new validations based on vibrational density of states (vDOS). vDOS is a particularly stringent test of the potential energy surface, and it captures the dynamics of the C-C and the C-H bonds. The MLP has good agreement with PBE across a wide range of relevant conditions and for different C/H compositions. These vDOS data are in the SI (Sec IV).

4. What are the statistical errors of the chemical potentials calculated here, considering the statistical features of machine learning? I suspect that the "liquid-liquid phase separation" may not exist because of the statistical uncertainty.

AUTHORS:

The statistical errors from the MD simulations using one MLP were propagated to obtain the statistical errors in the chemical potential, which are indicated using the error bars in Fig.3(a-f). The errors are quite small, and the LLPT conclusion is thus robust.

On the other hand, the errors introduced by the MLP are much harder to estimate directly. However, we have done the following to ensure that the conclusion on the liquid-liquid phase separation is robust:

- We have trained the MLP using a combination of different strategies, such as taking configurations from previous training sets of pure H and C, AIMD, random structure searches, and an iterative, active-learning procedure. The training set contains very diverse C/H structures with various atomic ratios. We further benchmarked the MLP using different properties including equations of states, radial distribution functions, vDOS, and diffusivities of various C/H compositions. The MLP has shown good accuracy at the relevant thermodynamic conditions.
- Importantly, at the temperature and pressure conditions where PS1 and PS2 happens, the RDFs of the MLP MD and the DFT MD from NVT simulations of small system size agree well. From the Kirkwood-Buff (KB) relationship, the RDFs of a two-component bulk system determine the chemical potential derivative $dmu/d\ln(x)$ (See Eqn.3). It is not possible to obtain $dmu/d\ln(x)$ from the KB relationship using the RDFs because of the finite size effects in the small NVT simulations, but the good agreement between the MLP and the DFT RDFs suggest that the MLP is able to accurately capture $dmu/d\ln(x)$, and thus the chemical potentials at different x .

- One way to gauge the statistical errors of ML models is to use the committee model (e.g. <https://aip.scitation.org/doi/full/10.1063/5.0036522>). This is essentially fitting multiple MLPs using different splits of training sets or random initialization, and looking at the spread of the computed physical quantities from MD simulations employing the different MLPs. For this study, we have constructed two versions of the MLPs (MLP-V1 and MLP-V2) using different training sets, and the MLP-V2 has better accuracy for structures with the low C concentrations. The two MLPs have very similar predictions on the phase separations. The phase diagram in Fig.3.g is based on MLP-V2, and we show the phase diagram based on MLP-V1 below. The regions for PS1 and PS2 are similar in both versions. This attests the robustness of our conclusions on the PS.

5. Does μ_{mixture} (free energy as in Fig. 3 c,f) include the mixing entropy in the calculations? How?

AUTHORS:

Yes, μ_{mixture} (chemical potential per atom in the C/H mixture as in Fig. 3 c,f) includes the factors that enter the chemical potential: the mixing entropy and enthalpy.

The mixing entropy is automatically accounted for using the S0 method. The detailed derivation can be found in Cheng 2022 (Ref. 42, <https://aip.scitation.org/doi/10.1063/5.0107059>).

To see how the mixing entropy enters the calculation, The Eqn.19 in Ref.42 expresses the chemical potential of a component A in the A/B mixture as

$$\mu_A(c_A) = \mu_A^0 + k_B T \ln\left(\frac{c_A}{c_A^0}\right) + k_B T \int_{\ln c_A^0}^{\ln c_A} d \ln(c_A) \left[\frac{1}{S_{AA}^0 - S_{AB}^0 \sqrt{c_A/c_B}} - 1 \right],$$

where c_A is the concentration. The first two terms on the right-hand side of the equation give the ideal-mixture chemical potential μ_A^{id} and the third term is the excess chemical potential μ_A^{ex} . The ideal-mixture chemical potential μ_A^{id} part comes from the ideal entropy of mixing. The chemical potential of the component B can be obtained analogously. As $\mu_{mixture} = x_A \mu_A + x_B \mu_B$, and it is easy to verify that the ideal part of $\mu_{mixture}$ account for the mixing entropy. We have updated the citation for Ref.42, as the paper is now published, as well as expanded the explanation for the S0 method in the main text.

6. The coexistence method was used to calculate the chemical potentials, and the authors said it agrees with the S0 method. However, I can see some differences as shown in Fig. 3. Considering this, is the linear fitting for the regions of liquid-liquid phase separation too arbitrary?

AUTHORS:

We used the S0 method to calculate the chemical potentials, and all the subsequent analysis is based on the values from the S0 method.

The coexistence method is what we performed in addition just for extra validations. Generally speaking, the coexistence method has a number of shortcomings: finite size effects and ergodicity issues related to the explicit interface; carbon concentration can vary in the liquid region of the simulation box; not efficient or feasible for low carbon concentrations. Overall, the statistical and systematic errors of the chemical potentials from the coexistence method are much larger compared to the S0 method. As such, all the conclusions regarding the liquid-liquid phase separation are based on the results from the S0 method only. The statistical errors in the chemical potentials from the S0 method are indicated using the error bars in Fig.3(b,c,e,f). The signatures of the phase separation (a plateau in the $\mu_c(x^c)$ in Fig.3(b,e) and the linear region in $\mu_{mixture}(x^c)$ in Fig.3(c,f)) are significant.

7. What is the highest nucleation rate in Fig. 1d? Is it possible to directly simulate the nucleation event at the highest nucleation rate? Or some enhanced method without predefined collective variables?

AUTHORS:

The nucleation rate can reach about $10^{25} / \text{m}^3\text{s}$ at about 20% of undercooling. We updated the color scale of Fig.1d to make this clearer.

Such high nucleation rates suggest that one can perform direct MD to sample the nucleation events at these conditions. To verify this, we performed a set of such MD simulations at (T=5000K, P=100 GPa), (T=5500K, P=200GPa), and (T=6000K, P=400GPa) and indeed observe nucleation followed by the solidification of the whole simulation box within the simulation time of 1ns. Below we show the snapshots of atomic positions during the MD run at (T=5000K, P=100 GPa).

Blue atoms are identified as having cubic diamond local structures, and the turquoise ones have hexagonal diamond structures. We did not directly observe any intermediate structures during the nucleation.

We have added the new MD simulation results to SI (Sec.V G), and discussions in the main text of the paper.

8. How does the production rate of diamonds change with the carbon ratio? It would be very interesting to find the maximum production rate.

AUTHORS:

The production rate of diamonds will likely go lower at a lower carbon fraction. The maximum production rate is probably when the system is pure carbon. This is because:

1) The chemical potential of C in the liquid phase decreases at lower carbon fractions, and the chemical potential is the thermodynamic driving force in diamond formations. Meanwhile, interfacial free energy between two phases is often lower when the structures of the two phases are similar, and vice versa. The interfacial free energy between diamond and the surrounding liquid thus may go higher with lower carbon concentration. A lower thermodynamic driving force and a (likely) higher surface energy imply that the kinetic barrier of nucleation will go up at lower carbon fractions.

(2) The kinetic prefactor of nucleation will likely be lower at lower carbon fractions, as carbon atoms in dilute solutions have to diffuse farther and farther on average to attach to the nuclei. We expanded the discussion to explain how carbon fractions change the nucleation rate. We have added these in the discussion section of the manuscript.

9. How would the liquid-liquid phase separation of C/H mixtures affect the diamond formation?

AUTHORS:

The liquid-liquid phase separations of C/H mixtures will enhance diamond formation:

- For PS1, the C/H mixture with a C atomic fraction in between will phase separate into two liquids with x_C of about 0.25 and 0.35. For the two liquid phases, their carbon atoms have the same chemical potential, but their interfacial free energies with diamond are different. Diamond will thus prefer to nucleate from the liquid phase with the lower interfacial free energy.

- For PS2, the liquid will separate into a phase with just hydrogen and almost no carbon, and another carbon-rich phase. In this zone, there is always thermodynamic driving force to form diamond from the carbon-rich phase. This means, PS2 greatly enhances the diamond formation at low carbon fractions.

We expanded the discussion to include these points.

10. "We show example snapshots of such phase separated configurations collected from the MD simulations in the Supplementary Information." It is better to show some videos for the phase separation.

AUTHORS:

This is a good idea. We have prepared a video and included it in the SI.

11. No unit in the legend of Figure 1c.

AUTHORS:

We have now added this. n_s is the number of atoms in the solid nucleus.

REVIEWER COMMENTS

Reviewer #2 (Remarks to the Author):

I thank the authors for their thorough replies.

I am satisfied by their replies to points 3 to 6 of my first review.

However, I would like to come back to points 1 and 2.

- Point 1 was on the accuracy of the DFT model, including the core electrons treatment and the choice of using the Born-Oppenheimer approximation. I would strongly suggest that an extended analysis, essentially mirroring the author's reply to my point 1, appeared in the method section of the manuscript. It is vital (i.e., it increases rather than hampers the quality of a paper) to identify the limitations of a model, and such description should not be buried into the SI.

- Point 2 was on the accuracy of the MLP compared to DFT. I am perfectly aware that the present study does not directly concern diamond-graphite transition, liquid-liquid transition(s), or gas phase hydrocarbons, but, is it so clear that e.g. an accurate modeling of torsional barriers around single, double, and conjugate bonds (including recognizing when a bond is single, etc.) in all circumstances is not an important aspect when modeling the phase boundaries and kinetics of compressed, high-temperature hydrocarbons? Checking controlled/known morphologies that are on explicitly part of the training set is only good practice Besides, fragments of these morphologies might (or even should) be contained in the disordered phases used for training the MLP. Similarly to point 1, describing (possible) limitations enhances the value of the paper, which already presents a compelling study of model hydrocarbons, with possible planetary-physics implications. As mentioned in my previous reply, the results are as good as the interaction model (both DFT and MLP built on top of it) and in the revised version of the manuscript I am afraid I still do not see that the accuracy of the interaction model is thoroughly discussed.

Reviewer #3 (Remarks to the Author):

I would like to thank the authors for answering my questions partly. I will consider to recommend this manuscript if the authors address my following comments properly.

1. "In principle, it may also be possible to compute the nucleation rate in the C/H mixture using the same framework, although the simulations will be very expensive considering the slower kinetic factor and to perform simulations at various C fractions."

I am glad that the authors in fact understand our concerns. The authors overclaimed their findings. The title of this work is "Diamond formation from hydrocarbon mixtures in planets", but when simulating the diamond formation at the atomistic scale, the authors started from pure carbon instead of hydrocarbons, which totally ignores the effects of hydrogen.

Why authors did not use MLPs to directly simulate the diamond formation in the C/H mixtures? MLPs are much more efficient than DFT. Metadynamics can overcome the slower kinetic factor, as shown by the authors. For one C fraction, the computational expense is similar to the current pure carbon simulation, so it is at least doable for one C fraction. I recommend to study two or three C fractions, and then interpolate, which is much better than extrapolating "the pure carbon case to the C/H mixture", "based on the ideal solution assumption".

2. "If there was a two-step transition during the nucleation process, the chosen CV (based on the number of diamond-like atoms) may not capture all the transition states. If that had happened, in metadynamics simulations using this CV, one would see a lot of hysteresis and the free energy profile will be very difficult to converge."

This is not correct. The intermediate state may not appear in the metadynamics simulation at all (without showing hysteresis) if the chosen CVs didn't pass the intermediate state. Because the intermediate state usually reduces the free energy barrier for the nucleus formation, I guess the free

energy barrier is overestimated.

3. The possible two-step nucleation mechanism is one example that the classical nucleation theory doesn't work. The authors calculated the nucleation rate based on many assumptions and linear fitting. Thus, I doubt the reliability of the calculated nucleation rate. Because the authors can directly simulate the diamond nucleation, I suggest them to calculate the nucleation rate directly [e.g., JPCB 112, 11060–11063 (2008); J. Am. Chem. Soc. 2015, 137, 42, 13658–13669].

4. The authors didn't validate their MLPs using the bond (C-H and C-C) lifetime as suggested before. The bond lifetime is very important for the hydrocarbon and diamond reactions. VDOS is not the bond lifetime.

5. "The ground truth in this study is density functional theory (DFT) within the PBE approximation to the exchange-correlation potential and within the Mermin finite-temperature formalism."

This is misleading. In this work, the authors did not apply "Mermin finite-temperature formalism". The temperature of electrons in this study is always 0 K. Considering the highest MD temperature in this study reaches 8000 K, should we seriously consider the temperature of electrons, ie., within Mermin's finite-temperature formalism?

We thank the referees for the comments. We have made a number of changes to the manuscript in response to their comments, and have highlighted these in blue in the revised version of the text. In what follows, we respond to each of the points raised by the referees. We hope we have addressed all the questions from the referees.

Reviewer #2 (Remarks to the Author):

I thank the authors for their thorough replies.

I am satisfied by their replies to points 3 to 6 of my first review.

However, I would like to come back to points 1 and 2.

- Point 1 was on the accuracy of the DFT model, including the core electrons treatment and the choice of using the Born-Oppenheimer approximation. I would strongly suggest that an extended analysis, essentially mirroring the author's reply to my point 1, appeared in the method section of the manuscript. It is vital (i.e., it increases rather than hampers the quality of a paper) to identify the limitations of a model, and such description should not be buried into the SI.

Authors:

We have now explicitly included discussions about the accuracy of the DFT model in the method section of the manuscript.

- Point 2 was on the accuracy of the MLP compared to DFT. I am perfectly aware that the present study does not directly concern diamond-graphite transition, liquid-liquid transition(s), or gas phase hydrocarbons, but, is it so clear that e.g. an accurate modeling of torsional barriers around single, double, and conjugate bonds (including recognizing when a bond is single, etc.) in all circumstances is not an important aspect when modeling the phase boundaries and kinetics of compressed, high-temperature hydrocarbons? Checking controlled/known morphologies that are on explicitly part of the training set is only good practice Besides, fragments of these morphologies might (or even should) be contained in the disordered phases used for training the MLP. Similarly to point 1, describing (possible) limitations enhances the value of the paper, which already presents a compelling study of model hydrocarbons, with possible planetary-physics implications. As mentioned in my previous reply, the results are as good as the interaction model (both DFT and MLP built on top of it) and in the revised version of the manuscript I am afraid I still do not see that the accuracy of the interaction model is thoroughly discussed.

Authors:

We have now explicitly included in the method section that the current MLP has been extensively benchmarked for high-pressure liquid hydrogen, diamond/liquid carbon, and C/H mixtures, but not for low-pressure carbon phases, diamond-graphite transitions or gas phase hydrocarbons.

Reviewer #3 (Remarks to the Author):

I would like to thank the authors for answering my questions partly. I will consider to recommend this manuscript if the authors address my following comments properly.

1. "In principle, it may also be possible to compute the nucleation rate in the C/H mixture using the same framework, although the simulations will be very expensive considering the slower kinetic factor and to perform simulations at various C fractions."

I am glad that the authors in fact understand our concerns. The authors overclaimed their findings. The title of this work is "Diamond formation from hydrocarbon mixtures in planets", but when simulating the diamond formation at the atomistic scale, the authors started from pure carbon instead of hydrocarbons, which totally ignores the effects of hydrogen.

Why authors did not use MLPs to directly simulate the diamond formation in the C/H mixtures? MLPs are much more efficient than DFT. Metadynamics can overcome the slower kinetic factor, as shown by the authors. For one C fraction, the computational expense is similar to the current pure carbon simulation, so it is at least doable for one C fraction. I recommend to study two or three C fractions, and then interpolate, which is much better than extrapolating "the pure carbon case to the C/H mixture", "based on the ideal solution assumption".

Authors:

We agree that diamond nucleation from C/H mixtures is very interesting. However, the focus of the current paper is on the thermodynamics of diamond formation from the mixtures, i.e. whether it is thermodynamically possible to form diamond at different P,T conditions and carbon concentrations. We have *not* extrapolated "the pure carbon case to the C/H mixture" "based on the ideal solution assumption". Instead, we accurately computed the chemical potentials of carbon in the C/H mixtures. We have edited the abstract to highlight these points.

The referee is right that our MLP can be used to directly simulate the diamond nucleation process in the C/H mixture, using a similar procedure we did for the pure carbon case. Such simulations are still quite expensive even using the MLP because one needs to properly converge the free energy profiles and kinetics of nucleation will be slower compared to the pure carbon case. However, we think this is beyond the scope of the current paper, particularly considering the breadth, the technical difficulties and the implications of the results already included in the manuscript.

2. "If there was a two-step transition during the nucleation process, the chosen CV (based on the number of diamond-like atoms) may not capture all the transition states. If that had happened, in metadynamics simulations using this CV, one would see a lot of hysteresis and the free energy profile will be very difficult to converge."

This is not correct. The intermediate state may not appear in the metadynamics simulation at all (without showing hysteresis) if the chosen CVs didn't pass the intermediate state. Because the

intermediate state usually reduces the free energy barrier for the nucleus formation, I guess the free energy barrier is overestimated.

Authors:

For a general case it is true that certain states will not be sampled if relevant CVs are not included in the enhanced sampling simulations. However, in the case of a two-step nucleation, if the number of the atoms in crystal-like environments (which is the CV we used) is still highly correlated with the second CV (the size of the intermediate phase), or if the nucleation barrier of the intermediate phase is low, one can still explore the two-step process from metadynamics simulations, albeit with hysteresis. For example, if the intermediate phase is also crystalline, (e.g. <https://www.pnas.org/doi/full/10.1073/pnas.2113059119>), it will be sampled using the current setup of the one CV.

Moreover, in the current case we did not observe any sign of the two-step transition in direct MD simulations at high undercooling, as we explained in the last revision. The same system has been studied before (e.g. Ref.25), and no two-step transition was observed either.

3. The possible two-step nucleation mechanism is one example that the classical nucleation theory doesn't work. The authors calculated the nucleation rate based on many assumptions and linear fitting. Thus, I doubt the reliability of the calculated nucleation rate. Because the authors can directly simulate the diamond nucleation, I suggest them to calculate the nucleation rate directly [e.g., JPCB 112, 11060–11063 (2008); J. Am. Chem. Soc. 2015, 137, 42, 13658–13669].

Authors:

We have indeed computed the diamond nucleation rate from liquid carbon directly at a number of conditions. In the figure below we show the free energy profile of nucleation computed from metadynamics simulations. In many conditions, the activation barrier of nucleation (G^*) was directly sampled, and the nucleation rate $J=k*\exp[-G^*/kBT]$ can be obtained without any assumption from classical nucleation theory or linear fitting.

Free energy profiles of diamond nucleation from liquid carbon at various conditions

4. The authors didn't validate their MLPs using the bond (C-H and C-C) lifetime as suggested before. The bond lifetime is very important for the hydrocarbon and diamond reactions. VDOS is not the bond lifetime.

Authors:

We have now added the additional validation of the MLPs based on the bond (C-H and C-C) lifetime as suggested. As shown below and in the SI, the MLP reproduces the bond lifetimes

extremely well. This, together with VDOS, provides a powerful demonstration that the MLP can capture the dynamics of the system very well.

Bond lifetimes from NVT simulations using PBE DFT and MLP.

5. “The ground truth in this study is density functional theory (DFT) within the PBE approximation to the exchange-correlation potential and within the Mermin finite-temperature formalism.”

This is misleading. In this work, the authors did not apply “Mermin finite-temperature formalism”. The temperature of electrons in this study is always 0 K. Considering the highest MD temperature in this study reaches 8000 K, should we seriously consider the temperature of electrons, i.e., within Mermin’s finite-temperature formalism?

Authors:

We have indeed applied the Mermin finite-temperature formalism. It is not correct that “the temperature of electrons in this study is always 0 K”. The ML potentials were trained on structures collected from MD simulations at finite temperatures, as well as low-temperature structures e.g. from crystal structure predictions. The forces and energies of high-temperature structures were computed using PBE and within the Mermin finite-temperature formalism with a

Fermi smearing (electron temperature) set to the average ionic temperature. Moreover, in the SI (Sec I B), we have extensively tested the influence of the Fermi smearing.

We now have added a sentence in the method to explain this:

“To take into account the impact of the thermal excitation of the electronic subsystem, we set the electronic temperature equal to the average ionic temperature during the DFT MD calculations as well as in the reference calculations used to train and test the MLP.”

REVIEWER COMMENTS

Reviewers' comments:

Reviewer #3 (Remarks to the Author):

I am very disappointed by the authors' reply. They deliberately avoided all the major questions here, which is a waste of time. The manuscript is not technically sound in methodology and analysis.

(1) "the referee is right that our MLP can be used to directly simulate the diamond nucleation process in the C/H mixture, using a similar procedure we did for the pure carbon case...", but "we think this is beyond the scope of the current paper..."

The title of this manuscript is "Diamond formation from hydrocarbon mixtures in planets". How come that "diamond formation from C/H mixtures" is beyond the scope? The MLPs together with metadynamics and extremely high temperatures makes the simulations absolutely feasible, but the authors refused again and again, so the only true reason is that their MLPs cannot describe the kinetic process from the C/H mixtures to diamonds. See my further comment about the quality of their MLPs below.

(2) "As shown below and in the SI, the MLP reproduces the bond lifetimes extremely well. "

I do not know how the authors defined "extremely well", but it is very obvious that the two figures on the left panel are not the same, even though they showed very crude comparison. For example, between 0.8 and 1.25 g/cc, the C-H bond life time predicted by MLPs is apparently longer than that from DFT.

Their MLPs are poorly constructed and not reliable. In the SI they wrote "For training the MLP V1, the resulting root mean squared errors (RMSE) of the energies in the training and test sets are 43 meV/atom and 42 meV/atom, respectively, and the RMSE values of the forces in the training and test sets are 865 meV/Å and 767 meV/Å, respectively. For training the MLP V2, the RMSE of the energies in the training and test sets are 42 meV/atom and 45 meV/atom, respectively, and the RMSE values of the forces in the training and test sets are 922 meV/Å and 800 meV/Å, respectively. "

The RMSEs in either training or test sets are way too large. In the authors' previous work (PNAS 116 (4) 1110, 2019), RMSE is only ≈ 7 meV/H₂O in energies and 120 meV/Å in forces for the test set. For a similar machine learning study of methane (Nat. Commun. 11, 5713 (2020)), the RMSE in forces is only 300 meV/Å.

(3) "We have indeed computed the diamond nucleation rate from liquid carbon directly at a number of conditions." "the nucleation rate $J = k \exp[-G^*/kBT]$ can be obtained without any assumption from classical nucleation theory or linear fitting."

No, they did not. The authors refused to read the two references that I gave in the last round, and have no idea what the classical nucleation theory is. The equation of nucleation rate, $J = k \exp[-G^*/kBT]$, is assumed based on the classical nucleation theory.

(4) "We have indeed applied the Mermin finite-temperature formalism. It is not correct that "the temperature of electrons in this study is always 0 K"

The authors do not know the Mermin finite-temperature formalism. The authors used the PBE functional, which was designed at 0 K, so in this work, the exchange-correlation interactions of electrons, which play a key role in DFT, were basically simulated at 0 K.

We thank the referees for their careful reading of our manuscript. We have made a number of changes to the manuscript in response to their comments, and have highlighted these in blue in the revised version of the text. In what follows, we respond to each of the points raised by the referees.

Reviewer #1 (Remarks to the Author):

The manuscript by Chen et al. examines the carbon-hydrogen phase diagram at extreme pressure-temperature conditions with the focus on diamond nucleation and growth from carbon-hydrogen mixtures. The authors go beyond using standard first-principles computational methods, which remain limited to relatively small system sizes and potentially failing to reveal real chemical/physical behavior. Among the important and novel findings of this study is the pressure-temperature space where any carbon-hydrogen fluid is thermodynamically driven to crystallize diamond (termed 'depletion zone'). The finding of 'depletion zone' potentially resolves the long-standing question about the apparently different luminosity of Uranus and Neptune, albeit these two planets being largely similar in many other physical parameters.

Overall, the manuscript is clearly written and understandable to a reader without comprehensive background in first-principles computations (such as myself). I recommend this work for publication in Nature Communications with the caveat that its technical details are approved by another reviewer in possession of computational expertise. Below I provide my questions and comments that together amount to a minor revision.

AUTHORS:

We thank the Referee for examining our work and giving a positive assessment.

--- Is it possible to make a quantitative estimate of the diamond production rate in the interiors of Uranus and Neptune in or above the 'depletion zone' given their expected carbon contents? If yes, can this rate be extrapolated in the past and compared to the overall carbon budget of these planets.

AUTHORS:

In principle it is possible to quantitatively estimate diamond formation in the planetary interiors based on our computed thermodynamic phase diagram (Fig.3g). In practice, this requires an accurate planetary interior model that contains pressure, temperature and hydrogen/carbon concentrations at different depths of the planets. Such a model should account for compositional gradients at different P,T conditions. Such a model is not currently available, partly because of the uncertainty in the observations/measurements of the planets, and partly because certain physical processes such as possible liquid-liquid phase separations and diamond formation are missing from the state-of-the-art planetary models. Extrapolating to the past is even more difficult because there is more uncertainty about the evolution of the planets.

--- 'In our chemical bond analysis, a C-C bond is identified whenever the distance between a pair of carbon atoms is within 1.6 Å, and a C-H bond is defined using a cutoff of 1.14 Å. These

cutoffs correspond to the first neighbor shells of the C-C and C-H radial distribution functions, respectively.'

A typical C-C bond in hydrocarbons is <1.6 Angstrom at ambient conditions and thus must be smaller at high pressure. Could you please clarify the choice of the 1.6 A cutoff (instead of, say, a variable or a smaller value). Please, also comment on the possible presence of carbon-carbon double and triple bonds in the simulations and how these affect the findings of the present work.

AUTHORS:

The 1.6 A cutoff here is selected based on the first neighbor shell of the C-C radial distribution functions (RDFs), i.e. the first minima in the RDFs. As shown in the RDFs of CH₄ below, this is a feature that is quite consistent across a wide temperature-pressure range. Moreover, as the Referee correctly pointed out, a typical C-C bond in hydrocarbons is less than 1.6 Angstrom, so having a cutoff that is 1.6 A ensures that thermal fluctuations of a C-C bond will not cause the bond to be misidentified as broken.

Due to the high energy of the systems at the high-pressure high-temperature conditions, the unsaturated carbon bonds are unlikely to exist.

We have added these explanations in the main text and the SI (Sec. IV D) of the paper.

--- Fig. 1 (c) – x-axis (n_s) needs a scale.

AUTHORS:

We have now added this. n_s is the number of atoms in the solid nucleus.

--- 'DFT studies coupled with harmonic approximations have predicted a variety of hydrocarbon crystals to be stable at $P \leq 300$ GPa [18–22], but these studies are restricted to low temperatures as the melting lines of hydrogen and methane are below 1000 K and 2000 K, respectively, while harmonic approximations break down completely for these liquids' In my opinion, references are in order for the melting curves of hydrogen and methane.

AUTHORS:

We have now added the references for the melting curves of hydrogen and methane.

Reviewer #2 (Remarks to the Author):

The manuscript by Cheng and co-authors presents a thorough computational study, which combines thermodynamics and kinetics, of diamond formation from pure carbon and hydrocarbons, up to 600 GPa and 8000K.

The study is very thorough, leveraging on methods that are standard in both computational statistical mechanics and machine-learning potential-energy-surface (PES) fitting. The only non-standard method is the recently introduced (by the first author of this manuscript) "S0" method. The merit of the manuscript is to integrate the study of pure-carbon with hydrocarbons in order to offer an updated answer to the long-standing "diamond in the sky?" question. The crucial step forward is to go beyond the ideal C,H mixture model assumed in the past. If the atomistic interaction model used in this work is reliable, then the results are credible and important.

My criticism, below, is therefore focused on both the ab initio modeling choice and the MLP trained on top o it.

AUTHORS:

We thank the Referee for carefully examining our work. We have added additional validations of the MLP in the revised version of the paper.

1 - How much realistic is Born-Oppenhemier MD via DFT for liquid C at the temperatures and pressures here studied? Liquid C must be metallic at those conditions, and the high density suggests also that an approximate treatment of core-electrons such as PAW (as opposed to an all-electron, full potential treatment) might be problematic. This is a crucial validation point as the comparison between the here presented results with experiments (in particular Kraus and Hartley) does not look consistent. I am ready to trust more the theory compared to very difficult experiments, IF the theory's scope is challenged and subsequently validated. Since there is no methodological advance for this manuscript, then everything is played at the level of predictions' reliability.

AUTHORS:

The Referee asked about the reliability of a 4 electron PAW potential for carbon at high pressure. We respond thusly:

Our DFT calculations are performed with the “hard” version of the PAW potentials. The accuracy of the carbon PAW potentials at high pressure has been the object of a study by Benedict et al. [Benedict et al. PRB 224109 (2014)], these authors reported that the 4 electron hard PAW from the VASP library compared well with a 6 electron (all-electron) PAW up to densities of 60 g/cc, corresponding to carbon-carbon distance of 1.3 atomic units (0.7 Angstroms). This is well beyond the range of densities encountered in this study.

To clarify this point we added this paragraph to the SI in section I:

“These consist of the hard version of the PAW potentials. The accuracy of this carbon PAW potential at high pressure has been the object of a study by Benedict et al. [Benedict et al. PRB 224109 (2014)], these authors reported that the 4 electron hard PAW from the VASP library compared well with a 6 electron (all-electron) PAW potential up to densities of 60 g/cc, corresponding to carbon-carbon distance of 1.3 atomic units (0.7 Angstroms).”

The referee also asked about the realtiness of the Born-Oppenheimer approximation in metallic carbon. To this we respond:

The ground truth in this study is density functional theory (DFT) within the PBE approximation to the exchange-correlation potential and within the Mermin finite-temperature formalism. Born-Oppenheimer DFT Molecular Dynamics is the workhorse of high pressure equation-of-state calculations and has shown good agreement with several experiments on hydrocarbons and other systems [Knudson et al. JAP **129**, 210904 (2021), Millot et al. Nature Physics **14**, 297 (2018), Kim et al. PRL **129**, 015701 (2022)] for measured thermodynamic properties in particular for Hugoniot curves in pressure-density space and in pressure-temperature space when the temperature is inferred with pyrometry.

Regarding the comparison between our computed melting curve with Eggert et al. [Nature Physics **6**, 40 (2010)], the agreement is good (capturing the reentrant behavior of the diamond melting line at high pressure). The experiments have uncertainties and now we have plotted the uncertainties in Fig.1.a. There is an offset in temperature between the BOMD calculations of the melting line and the experimental melting line and one possible explanation is systematic uncertainties in their temperature estimates. A recent set of experiments (as yet unpublished) that are in closer agreement with our DFT BOMD melting line supports this explanation [Millot, private communication].

For the diamond formation in C/H mixtures, experiments are scattered and contradicting with each other. There, possible reasons for the disagreement are kinetics effects and temperature inference for the reported thermodynamic states (PT points). In particular, the temperatures reported by Hartley et al. were inferred from an equation of state (SESAME 7171) that was used outside the range of pressure and temperature where it was validated ($P < 50$ GPa) [Knudson et

al. JAP **129**, 210904 (2021)] possibly leading to large systematic uncertainties in their temperature inference.

We added this paragraph to the SI in section II:

“Born-Oppenheimer DFT Molecular Dynamics is the workhorse of high pressure equation-of-state calculations and has shown good agreement with several experiments on hydrocarbons and other systems [Knudson et al. JAP **129**, 210904 (2021), Millot et al. Nature Physics **14**, 297 (2018), Kim et al. PRL **129**, 015701 (2022)] for measured thermodynamic properties in particular for Hugoniot curves in pressure-density space and in pressure-temperature space when the temperature is estimated with pyrometry.”

2 - How reliable is the MLP itself, compared to the reference DFT? I see, as it is typical for MLPs, a (very thorough, in particular in the impressive SI) statistical validation of the MLP, plus a report on random structure search. This is nice, but phase coexistence and nucleation kinetics depend on the accurate description of the subtle chemistry of C, with its bonding flexibility. What about formation enthalpy and vibrational properties (spectra) of small hydrocarbon molecules, including aromatic ones? What about the diamond to graphite transition barrier and the torsional energy of single, aromatic, double bonds? The latter point was for instance crucial for LCBOP when discussing the liquid-liquid phase transition. Obviously, such validation is more compelling if such molecules and structures are not explicit part of the training set.

AUTHORS:

The Referee asked about the reliability of our MLP, particularly regarding the subtle chemistry of C. We respond as the following:

- The training set of our MLP explicitly includes a large variety of carbon structures/molecules with single, aromatic, double bonds: cubic/hexagonal diamond, graphite, graphane, carbon nanotubes, fullerenes, amorphous carbon, and defected structures. Indeed, all of the training structures of the GAP-20 model for carbon (Ref. 57, <https://aip.scitation.org/doi/pdf/10.1063/5.0005084>) are included in the training of our MLP. The GAP-20 carbon MLP is regarded to well-capture the chemistry of carbon, because it has been thoroughly benchmarked using metrics such as formation energies, radial distribution functions and the torsional energies of carbon bonds. Therefore, the training set is comprehensive enough to capture the chemistry of C.
- Although the MLP is able to describe the enthalpies of diamond and graphite, we do not study the diamond-graphite transition in the present paper, and we do not discuss the liquid-liquid phase transition in liquid carbon. As such, these topics, while extremely interesting, are out-of-scope for the present paper.
- Our MLP is for bulk C/H mixtures at $10 \text{ GPa} \leq P \leq 600 \text{ GPa}$ and $2500 \text{ K} \leq T \leq 8000 \text{ K}$, and is not applicable for small hydrocarbon molecules in the gas phase. The gas-phase molecules are also not relevant for our study.
- In the SI, we have already included many validations of the MLP, based on equations of states, radial distribution functions, and diffusivities of various C/H compositions.
- Nevertheless, we have added new validations based on vibrational density of states (vDOS). vDOS is a particularly stringent test of the potential energy surface, and the

MLP has good agreement with PBE across a wide range of relevant conditions and for different C/H compositions.

To clarify these points, we have expanded the explanation of the training set in the Methods section and in the SI (Sec III A), and add vDOS to SI (Sec. IV D,E,F).

3 - Why a newly trained MLP is necessary at all, considering the extensive work presented by Rowe et al. (ref. 56)? I.e., why does one need Behler-Parrinello symmetry functions and the related NN machinery vs the SOAP-GAP solution? I would like to see some discussion of this. It is nice that, compared to the unsystematic times of empirical potentials when designing a training set and a training procedure was an art (quoting a famous paper by Donald Brenner), MLPs can be systematically trained by (in principle) everybody. However, I think one should justify why one more MLP needs to be created.

AUTHORS:

The new MLP that we trained for this study is absolutely necessary because (to the best of our knowledge) all the previous MLPs cannot describe the C/H mixture with variable compositions at high-pressures (10-600 GPa). The ones by Rowe et al. (GAP-20) is for pure carbon only, and only works for pressures less than about 50 GPa (liquid carbon of density 3.5 g/mL). We have highlighted these justifications in the manuscript.

4 - In this respect, is there any plan to provide to the future readers the full training set(s) and procedure (in terms of actual data and scripts) in a localized repository (matbench, NOMAD, etc.)? We are in the era of FAIR data: users need to be able to reproduce results without barriers in order to accelerate the development of models and understanding.

AUTHORS:

We agree completely. We have already uploaded all the data needed to reproduce the results, including training set, trained MLP and the input files to train the MLP in the GitHub repository: <https://github.com/BingqingCheng/highp-ch>

BingqingCheng / highp-ch Private

<> Code Issues Pull requests Actions Projects Security Insights Settings

main 1 branch 0 tags Go to file Add file Code

BingqingCheng forgot to add plumed.dat 4b31ab6 on 2 Aug 6 commits

gdr	added gdr	2 months ago
interface-pinning-diamond-liquidca...	add files	3 months ago
metad-diamond-in-liquidcarbon	forgot to add plumed.dat	2 months ago
mlp-md-example	add files	3 months ago
nnp	add new train set for V2	2 months ago
npt-CxHy	add files	3 months ago
scripts	add files	3 months ago
train-set	add new train set for V2	2 months ago
vasp-input	add vasp input	2 months ago
README.md	Initial commit	3 months ago

For now the repository is private, and we will make it public upon the acceptance of the manuscript.

In the past, we have always followed this practice, and made available our MLPs (with training sets) for water, superionic water, high-pressure hydrogen.

5 - I would like to understand the implications of Fig2b better. The 1-bond region suggests that on average one has C-C dimers, probably duly saturated by hydrogens. I suspect that C₂H_x is not the most probable species, though, but rather one observes a mixture of CH₄ and short chains. What is the distribution of C-chain length in the 1-bond region (e.g., at few selected state points)? Crucially, how to explain/characterize the narrow strip of 2-bonds liquid, separating the 1-bond and 3-bond liquids? Is it a phase transition? A first order one? Such boundary region seems even to end in a critical point around 500 GPa.

AUTHORS:

The Referee asked for more clarifications for the implications of Fig2b. In general, the C-C bond lifetime is short (less than about 1 ps), across the whole conditions of Fig.2(b-e), 10 GPa ≤ P ≤ 600 GPa and 2500 K ≤ T ≤ 7000 K. This means that hydrocarbon chains in the systems decompose and form quickly. The C/H mixture is better characterized as a liquid made of C and H atoms, rather than a polymer mixture. To reveal this further, in the figure below we show the distribution for the number of C-C bonds that each carbon atom has. It can be seen that, at all conditions considered, the distribution is rather broad than sharply peaked at a specific bond number. In other words, if on average each carbon atom has 1 bond, this means some C atoms have one bond, some have two bonds, some have zero bond, and so on. All these “bonds” are extremely transient.

The Referee also asked about the narrow strip of 2-bond liquid. Indeed, the original choice of the color scheme (matplotlib, cmap='terrain') in the Fig.2.b made the 2-bond region look particularly narrow because of the drastic color gradient at $n=2$. Now we have changed the color scheme (cmap='rainbow') to eliminate such effects.

Now it can be seen that the average number of C-C bonds varies gradually at the conditions considered here: the number of bonds are higher at higher pressures as well as lower temperatures. There is no first-order phase transition for this CH₄ system, judging from the average number of C-C bonds.

The 2-bond region gets narrower at low temperatures. This can be rationalized using the results from Fig.3(a,d): the CH₂ composition ($X_c=0.33$) is not thermodynamically favorable ($d\mu/dx_c \sim 0$), and it tends to undergo a liquid-liquid phase separation at lower temperatures (the PS1 mechanism). However, the analysis based on the chemical potentials of the C/H mixture presented in Fig.3 is much more rigorous and quantitative than the observation about the 2-bond region, so we refrain from drawing any direct conclusions from the latter.

We have expanded the SI (Sec V H) and the main text to incorporate the discussion above.

6 - I do not recognize the wording "fully dissociable" attribute to the MLP presented in the manuscript. I understand that there is a whole class of non-dissociable (normally called "non reactive", I believe, i.e.,) potentials where bonds do not break by construction, widely used in soft matter and biophysics. However, I am not keen on recognizing degrees of dissociability, such that the present potential is fully dissociable and others (e.g.?) only partially. So, my suggestion is that it is fine to point out that this MLP treats bond breaking and forming, but I would leave out the unclear "fully", unless I am convinced otherwise.

AUTHORS:

We have removed the word "fully".

I will be happy to reconsider the publication of this manuscript after the above points (in particular 1, 2, and 5).

Reviewer #3 (Remarks to the Author):

Cheng et. al. studied the diamond formation out of pure liquid carbon and the P-T phase boundaries using machine learning force field. I do not think that the manuscript reaches the publication level required by Nature Communications. Below are my comments:

AUTHORS:

We thank the Referee for taking the time to critically examine our work and provide constructive suggestions. We have revised the manuscript following the comments of the Referee.

1. The title is misleading. When talking about diamond formation from hydrocarbons, we would like to know both the kinetics and thermodynamics, e.g., how diamonds are generated out of hydrocarbons at the atomistic scale. But this work only provides P-T conditions for the possible existence of diamonds, so the work doesn't match my excitement and expectation when reading the title. Furthermore, many experiments studied C/H mixtures, and gave inconsistent P-T conditions for the diamond formation, as shown in the introduction. This work agrees with some experiments but not with others, which the authors attributed to "the kinetic effects". However, the authors did not give any evidence about kinetics to support this claim.

AUTHORS:

We are confused by this comment from the Referee. Indeed we studied both the thermodynamics and kinetics of diamond formation:

- Thermodynamics: Our motivation to study diamond formation from high-pressure hydrocarbons is to probe the existence of such processes in planets. Crucially, planets have billions of years to evolve and the C/H mixtures inside the planets have a long time to equilibrate, so thermodynamic equilibrium or quasi-thermodynamic equilibrium are good assumptions for the states of hydrocarbons in planets.

To provide the thermodynamic picture of diamond formation from hydrocarbons, we computed the thermodynamic driving force $\Delta\mu$ for the carbon to form the diamond phase from the C/H liquid mixture at a wide range of P-T conditions and with different carbon fractions. In the thermodynamic phase diagram (Fig.3.g), the phase boundary where $\Delta\mu = 0$ provides the P-T conditions for the possible existence of diamonds, taking into account all the thermal effects including the mixing entropy and the enthalpy of the C/H liquid mixture, as well as the thermal fluctuations of the diamond phase. This phase diagram shows under which conditions diamond formation is possible, and under which conditions it is impossible. These conditions can then be compared to the internal conditions of planets.

It is worth mentioning that, to the best of our knowledge, this is the first time that chemical potentials of mixtures are computed at the ab initio-accuracy level.

- Kinetics: We computed the kinetic nucleation rates for diamond formation in pure liquid carbon at a wide range of thermodynamic conditions. Such nucleation rates account for the activation barrier and the kinetic prefactor of nucleation. The nucleation rates can inform us that at a wide range (P,T) conditions whether liquid carbon will solidify into diamond or remain metastable during a given timescale.

In principle, it may also be possible to compute the nucleation rate in the C/H mixture using the same framework, although the simulations will be very expensive considering

the slower kinetic factor and to perform simulations at various C fractions. Alternatively, one can extrapolate the pure carbon case to the C/H mixture, e.g. Ghiringhelli et al. computed the nucleation rates at two state points using the LCBOP model (Ref 25, see Fig.1d), and then extrapolated the rates to other thermodynamic conditions and carbon fractions (based on the ideal solution assumption). In comparison, our computations are much more quantitative because the nucleation rates are based on MLPs with ab initio-accuracy and were computed at many conditions, and our chemical potentials of carbon were computed accurately.

Finally, It is worth noting that, using the machine learning potential that we constructed, it is possible to run dynamic compression simulations. However, the situation in the planets is closer to the thermodynamic equilibrium picture shown in Fig.3.g, than the dynamic compression simulations/experiments that live on the nanosecond timescale.

Regarding the disagreement with experiments, we speculate that the possible cause may be due to the kinetic effects in the rapid compression and/or the difficulty in the temperature estimation of these experiments. Here the kinetic effects refer to that these experiments are fast but the diamond nucleation may take longer time to happen. The temperature estimation of these experiments are difficult and can be inaccurate, because the temperatures are often not directly measured, but rather inferred from a specific equation-of-state model that may or may not be precise in a particular regime of pressure and temperature. In particular, the temperatures reported by Hartley et al. were inferred from an equation of state (SESAME 7171) that was used outside the range of pressure and temperature where it was validated ($P < 50$ GPa) [Knudson et al. JAP **129**] possibly leading to large systematic uncertainties in their temperature inference. Moreover, even the occurrence of diamond formation is nontrivial to verify for the reactions happening inside a diamond anvil cell. Due to all these complications, we can only speculate the cause of the disagreement. We have rewritten the discussion to make it clear the speculative nature of our explanation for the experimental results.

2. For the diamond nucleation out of pure liquid carbon, the authors used metadynamics to obtain the free energy barrier for the diamond formation. However, the collective variable (CV) is simply the number of atoms that have the diamond structure, which may not be a good assumption if the carbon nucleation has more than one step. In fact, the authors also mention the two-step nucleation mechanism on page 5, ref. 42. The possible intermediate structures may bring some problems when defining CVs to calculate energy barriers. Moreover, the intermediate structures can be related to non-classical nucleation mechanisms, which means that eq. 1 from the classical nucleation theory is problematic. Furthermore, the surface energy is "extrapolated to other conditions using a linear fit in both P and T ." Is it reasonable? With so many assumptions and approximations, the conclusions here are uncertain.

AUTHORS:

The Referee suggested that the choice of the CV may not be good if there is a two-step nucleation process. We respond as follows:

- We performed direct MD simulations to probe the nucleation process at deeply undercooled regime, and did not observe any sign of the two-step mechanism (see below).
- In our metadynamics simulations, the free energy profiles were easy to converge with little hysteresis, which suggests that the nucleation process here is a simple one-step process. If there was a two-step transition during the nucleation process, the chosen CV (based on the number of diamond-like atoms) may not capture all the transition states. If that had happened, in metadynamics simulations using this CV, one would see a lot of hysteresis and the free energy profile will be very difficult to converge. One would need another CV to run enhanced sampling simulations in the 2D space to properly converge the free energies. However, this is not the case here.

Regarding the extrapolation of the surface energy to obtain the nucleation barrier, we have the following justifications:

- The most relevant conditions for studying nucleation rates are between about 5% to 20% of undercooling, because at lower undercooling the nucleation rate is negligibly small and at higher undercooling nucleation will happen almost instantly so it is not particularly meaningful to compute the exact rates. For the relevant range of conditions, we have indeed run metadynamics simulations and computed the free energy profiles directly (Fig.S22). These conditions include: 25GPa, 50GPa, 100GPa, 200GPa, 300GPa, 400GPa, 500GPa, 600GPa, and at different undercoolings ranging between 8% and 20%. As such, for the relevant range we are indeed performing interpolation rather than extrapolation within this wide range of conditions. Moreover, in many of these simulations, the nucleation barriers were sampled directly.
- Interfacial free energies often have a linear dependence in temperature (e.g. Cahn Hilliard model <https://aip.scitation.org/doi/pdf/10.1063/1.1744102>). The pressure dependence is less clear but for the case of diamond-liquid carbon interface the pressure dependence was found to be rather weak anyways. So the linear fit in both P and T is reasonable. Moreover, Ghiringhelli et al. (Ref 25) extracted the surface energy at two state points (A=(30GPa, 3750K) and B=(85GPa, 5000K)), and then linearly extrapolated the surface energy to other thermodynamic conditions using the fraction of threefold coordinated carbon atoms. We also tried using this fraction in the linear fit, but did not find it to improve the quality of the fit, so we finally used the linear fit based on P and T .

We have expanded the manuscript and the SI to include the discussions above.

3. In Figs. S4 and S10, it seems that the radial distribution functions (RDFs) obtained from MLP do not match well with the DFT data at some conditions. Anything wrong with MLP? Furthermore, besides energies and RDFs, can the authors also validate the dynamics of C-C and C-H bonds, e.g., the bond lifetime, which are important for hydrocarbon/diamond reactions?

AUTHORS:

Figs. S4 compares the carbon-carbon radial distribution function of pure liquid carbon computed from NVT simulations using DFT and MLP. The discrepancy can be seen at 6000K, e.g. at $\rho=5.7, 6.1,$

6.5g/mL. This is because the liquid carbon at 6000K and density close to 6g/mL tends to solidify during the NVT simulations, and the affected conditions can be seen from the points of abnormally low PE in the figure below. This solidification happens at different timesteps during the NVT simulations, causing the MLP and DFT RDFs to look different.

Fig. S10 shows the C-C RDFs for CH₄. The initial configuration of the DFT simulations is methane molecules on a bcc lattice. In hindsight, this was not a good choice because the system can get stuck in this local minima during very short MD simulations: The bcc methane is not the stable phase and should become liquid at $T \geq 2000$ K (above the melting line). At $T \geq 4000$ K the bcc lattice always melts, but at $T = 3000$ K the system is not ergodic. At $T = 3000$ K, the difference between the MLP and the DFT RDFs for CH₄ is thus again due to the ergodicity problem and the different onset of phase transition. For example, at $T = 3000$ K, $\rho = 1.5$ g/mL, the final configuration is still solid.

Snapshots from DFT MD simulation of CH₄ at 3000K, $\rho = 1.5$ g/mL

The RDFs also differ significantly at 4.50 g/mL, but this corresponds to pressures higher than 1000GPa, which we do not consider in the present study.

For the other C/H compositions including CH₁₆, CH₈, CH₂, CH, C₂H, C₄H, the initial configurations were chosen to be a randomly generated structure, and such ergodicity problems did not appear.

To summarize, at relevant conditions, the difference in the RDFs in a few conditions is due to the ergodicity problem of short MD simulations. It is not related to the accuracy of the MLP. The reasons for the discrepancy was explained in the (very long) SI. However, we understand that many readers will not go through all the details in the SI, and the previous figures on the RDFs may be misleading. As such, in the revised SI, we kept the discussion of the discrepancy in the text, but removed the RDFs for liquid carbon at 6000K and the RDFs for CH₄ at 3000K as these results are not physically meaningful. We also removed the results for CH₄ for $\rho \geq 4.0 \text{ g/mL}$ (which corresponds to $P \geq 800 \text{ GPa}$), as such pressure conditions are not relevant for our study. We hope in this way the SI is more informative and a bit simpler.

Moreover, we have added new validations based on vibrational density of states (vDOS). vDOS is a particularly stringent test of the potential energy surface, and it captures the dynamics of the C-C and the C-H bonds. The MLP has good agreement with PBE across a wide range of relevant conditions and for different C/H compositions. These vDOS data are in the SI (Sec IV).

4. What are the statistical errors of the chemical potentials calculated here, considering the statistical features of machine learning? I suspect that the "liquid-liquid phase separation" may not exist because of the statistical uncertainty.

AUTHORS:

The statistical errors from the MD simulations using one MLP were propagated to obtain the statistical errors in the chemical potential, which are indicated using the error bars in Fig.3(a-f). The errors are quite small, and the LLPT conclusion is thus robust.

On the other hand, the errors introduced by the MLP are much harder to estimate directly. However, we have done the following to ensure that the conclusion on the liquid-liquid phase separation is robust:

- We have trained the MLP using a combination of different strategies, such as taking configurations from previous training sets of pure H and C, AIMD, random structure searches, and an iterative, active-learning procedure. The training set contains very diverse C/H structures with various atomic ratios. We further benchmarked the MLP using different properties including equations of states, radial distribution functions, vDOS, and diffusivities of various C/H compositions. The MLP has shown good accuracy at the relevant thermodynamic conditions.
- Importantly, at the temperature and pressure conditions where PS1 and PS2 happens, the RDFs of the MLP MD and the DFT MD from NVT simulations of small system size agree well. From the Kirkwood-Buff (KB) relationship, the RDFs of a two-component bulk system determine the chemical potential derivative $dmu/d\ln(x)$ (See Eqn.3). It is not possible to obtain $dmu/d\ln(x)$ from the KB relationship using the RDFs because of the finite size effects in the small NVT simulations, but the good agreement between the MLP and the DFT RDFs suggest that the MLP is able to accurately capture $dmu/d\ln(x)$, and thus the chemical potentials at different x .

- One way to gauge the statistical errors of ML models is to use the committee model (e.g. <https://aip.scitation.org/doi/full/10.1063/5.0036522>). This is essentially fitting multiple MLPs using different splits of training sets or random initialization, and looking at the spread of the computed physical quantities from MD simulations employing the different MLPs. For this study, we have constructed two versions of the MLPs (MLP-V1 and MLP-V2) using different training sets, and the MLP-V2 has better accuracy for structures with the low C concentrations. The two MLPs have very similar predictions on the phase separations. The phase diagram in Fig.3.g is based on MLP-V2, and we show the phase diagram based on MLP-V1 below. The regions for PS1 and PS2 are similar in both versions. This attests the robustness of our conclusions on the PS.

5. Does μ_{mixture} (free energy as in Fig. 3 c,f) include the mixing entropy in the calculations? How?

AUTHORS:

Yes, μ_{mixture} (chemical potential per atom in the C/H mixture as in Fig. 3 c,f) includes the factors that enter the chemical potential: the mixing entropy and enthalpy.

The mixing entropy is automatically accounted for using the S0 method. The detailed derivation can be found in Cheng 2022 (Ref. 42, <https://aip.scitation.org/doi/10.1063/5.0107059>).

To see how the mixing entropy enters the calculation, The Eqn.19 in Ref.42 expresses the chemical potential of a component A in the A/B mixture as

$$\mu_A(c_A) = \mu_A^0 + k_B T \ln\left(\frac{c_A}{c_A^0}\right) + k_B T \int_{\ln c_A^0}^{\ln c_A} d \ln(c_A) \left[\frac{1}{S_{AA}^0 - S_{AB}^0 \sqrt{c_A/c_B}} - 1 \right],$$

where c_A is the concentration. The first two terms on the right-hand side of the equation give the ideal-mixture chemical potential μ_A^{id} and the third term is the excess chemical potential μ_A^{ex} . The ideal-mixture chemical potential μ_A^{id} part comes from the ideal entropy of mixing. The chemical potential of the component B can be obtained analogously. As $\mu_{mixture} = x_A \mu_A + x_B \mu_B$, and it is easy to verify that the ideal part of $\mu_{mixture}$ account for the mixing entropy. We have updated the citation for Ref.42, as the paper is now published, as well as expanded the explanation for the S0 method in the main text.

6. The coexistence method was used to calculate the chemical potentials, and the authors said it agrees with the S0 method. However, I can see some differences as shown in Fig. 3. Considering this, is the linear fitting for the regions of liquid-liquid phase separation too arbitrary?

AUTHORS:

We used the S0 method to calculate the chemical potentials, and all the subsequent analysis is based on the values from the S0 method.

The coexistence method is what we performed in addition just for extra validations. Generally speaking, the coexistence method has a number of shortcomings: finite size effects and ergodicity issues related to the explicit interface; carbon concentration can vary in the liquid region of the simulation box; not efficient or feasible for low carbon concentrations. Overall, the statistical and systematic errors of the chemical potentials from the coexistence method are much larger compared to the S0 method. As such, all the conclusions regarding the liquid-liquid phase separation are based on the results from the S0 method only. The statistical errors in the chemical potentials from the S0 method are indicated using the error bars in Fig.3(b,c,e,f). The signatures of the phase separation (a plateau in the $\mu_c(x^c)$ in Fig.3(b,e) and the linear region in $\mu_{mixture}(x^c)$ in Fig.3(c,f)) are significant.

7. What is the highest nucleation rate in Fig. 1d? Is it possible to directly simulate the nucleation event at the highest nucleation rate? Or some enhanced method without predefined collective variables?

AUTHORS:

The nucleation rate can reach about $10^{25} / \text{m}^3\text{s}$ at about 20% of undercooling. We updated the color scale of Fig.1d to make this clearer.

Such high nucleation rates suggest that one can perform direct MD to sample the nucleation events at these conditions. To verify this, we performed a set of such MD simulations at $(T=5000\text{K}, P=100\text{ GPa})$, $(T=5500\text{K}, P=200\text{GPa})$, and $(T=6000\text{K}, P=400\text{GPa})$ and indeed observe nucleation followed by the solidification of the whole simulation box within the simulation time of 1ns. Below we show the snapshots of atomic positions during the MD run at $(T=5000\text{K}, P=100\text{ GPa})$.

Blue atoms are identified as having cubic diamond local structures, and the turquoise ones have hexagonal diamond structures. We did not directly observe any intermediate structures during the nucleation.

We have added the new MD simulation results to SI (Sec.V G), and discussions in the main text of the paper.

8. How does the production rate of diamonds change with the carbon ratio? It would be very interesting to find the maximum production rate.

AUTHORS:

The production rate of diamonds will likely go lower at a lower carbon fraction. The maximum production rate is probably when the system is pure carbon. This is because:

1) The chemical potential of C in the liquid phase decreases at lower carbon fractions, and the chemical potential is the thermodynamic driving force in diamond formations. Meanwhile, interfacial free energy between two phases is often lower when the structures of the two phases are similar, and vice versa. The interfacial free energy between diamond and the surrounding liquid thus may go higher with lower carbon concentration. A lower thermodynamic driving force and a (likely) higher surface energy imply that the kinetic barrier of nucleation will go up at lower carbon fractions.

(2) The kinetic prefactor of nucleation will likely be lower at lower carbon fractions, as carbon atoms in dilute solutions have to diffuse farther and farther on average to attach to the nuclei. We expanded the discussion to explain how carbon fractions change the nucleation rate. We have added these in the discussion section of the manuscript.

9. How would the liquid-liquid phase separation of C/H mixtures affect the diamond formation?

AUTHORS:

The liquid-liquid phase separations of C/H mixtures will enhance diamond formation:

- For PS1, the C/H mixture with a C atomic fraction in between will phase separate into two liquids with x_C of about 0.25 and 0.35. For the two liquid phases, their carbon atoms have the same chemical potential, but their interfacial free energies with diamond are different. Diamond will thus prefer to nucleate from the liquid phase with the lower interfacial free energy.

- For PS2, the liquid will separate into a phase with just hydrogen and almost no carbon, and another carbon-rich phase. In this zone, there is always thermodynamic driving force to form diamond from the carbon-rich phase. This means, PS2 greatly enhances the diamond formation at low carbon fractions.

We expanded the discussion to include these points.

10. "We show example snapshots of such phase separated configurations collected from the MD simulations in the Supplementary Information." It is better to show some videos for the phase separation.

AUTHORS:

This is a good idea. We have prepared a video and included it in the SI.

11. No unit in the legend of Figure 1c.

AUTHORS:

We have now added this. n_s is the number of atoms in the solid nucleus.

For this round of review, I will focus on the disagreements between the authors and referee #3, as asked by the editor. In particular, I will quote (excerpts from) and comment the document CH3-revision-3 (383671_3_art_file_7121085_rlwt2x.pdf within NCOMM's reviewing system).

Quoted ref. #3 comments are in black and authors' reply in blue. My comments are in green.

- Ref #3 - The title of this manuscript is "Diamond formation from hydrocarbon mixtures in planets". How come that "diamond formation from C/H mixtures" is beyond the scope?

My comment - The reviewer has a fair point here. I agree the title is overselling the results, given that only thermodynamics conditions are examined for CH mixtures, i.e., kinetic effects are not analysed. I strongly suggest this is made clear by modifying the title into "Thermodynamics conditions for ..." or something similar. I appreciate that the abstract has been already modified in this direction in the latest version.

- Authors - The referee insisted on studying the nucleation of diamond from hydrocarbons, but we think this is beyond the scope of the current paper, particularly considering the breadth, the technical difficulties and the implications of the results already included in the manuscript.

I agree that a thorough kinetic study for CH mixtures is more than a simple addition to the manuscript, which has its merits with the current content.

However, I see that there is an asymmetry in reporting the kinetic study only for the pure C, then leaving the reader on a "cliffhanger" on how the kinetic story ends. I understand the authors' argument about the comparison to a previous study on diamond nucleation with a different model potential, but wouldn't it make more sense to have a dedicate paper on the kinetics, for both pure C and CH mixtures?

This would also allow the possibility to look more carefully into collective variables. As Ref #3 hints at in their previous report, it has been shown in the past that size of the crystallite might not be always the best choice, or, in general one may need more than one CV, i.e., by adding info on the shape of the crystallite (to my knowledge, this was first shown by Moroni, Ten Wolde, Bolhuis in their 2005's PRL, but I am sure the authors know the fundamentals in the field).

- Moreover, planets have billions of years to evolve, so the thermodynamic picture of whether diamonds can form or not is perhaps more important than the transient mechanism of diamond nucleation.

This is an unscientific reply. As the authors know, nucleation rates in condensed matter may well span tens of orders of magnitude. One may get a nucleation rate that is much less than one event per planet per solar system, in a region where thermodynamics says the crystal is more stable. So, it is not true that thermodynamic conditions are more important than the transient mechanism. I would suggest the authors tell clearly that kinetics could matter, which was the whole point of Ghiringhelli, Valeriani, *et al.*, 2007, I understand.

- "As shown below and in the SI, the MLP reproduces the bond lifetimes extremely well." I do not know how the authors defined "extremely well", but it is very obvious that the two figures on the left panel are not the same, even though they showed very crude comparison. For example, between 0.8 and 1.25 g/cc, the C-H bond life time predicted by MLPs is apparently longer than that from DFT.

I partially agree with the referee, here. The plots, currently Fig. S14 in SI seem to show some possibly significant discrepancy between DFT and and the MLP. I think the chosen

layout of the plot is not very readable, with the color map. I suggest to have 5x2 plots (one per temperature and for both CC and CH bond lifetimes) with density on the x-axes and the lifetimes on the y-axes (plots can be merged if still legible).

This said, lifetimes might be more relevant for kinetics than thermodynamics, so as a recommendation for the future work on kinetics, the MLP might need some improvement. In short, I believe the authors are a bit overselling their MLP. But I am also confident that for a thermodynamic study the discrepancies are not crucial. As mentioned in earlier reports, I encourage the authors to be more open on the shortcomings, in the manuscript itself. I applaud the thorough DFT-MLP comparison reported in SI, but I would point out that, e.g., dynamic properties are only fairly good at low T and rho (Fig. SI3), that liquid methane is quite overstructured at low rho and higher T.

- Their MLPs are poorly constructed and not reliable. [...] The RMSEs in either training or test sets are way too large.

The referee's statements are too strong, in my opinion. The MLP was carefully trained – and tested! – and I believe that the errors are acceptable for the thermodynamic study

- The referee mentioned the training errors of other systems (low-pressure water and methane). However, the training error is only meaningful when comparing it to the energy span of the system.

Here, I disagree with the authors. The relative error, or error compared to the span of values of the trained quantity is the typical “ML experts” figure of merit, as one wants to know how better is the trained model than just predicting the average value of the quantity for every new data point. For a thermodynamic study, I would say that the errors have to be projected into reliability of the phase boundaries, e.g., in terms of kelvins and gigapascals. For instance, how reliable are the boundaries of the “depletion zone” in Fig. 3G, in terms of K and GPa?

- (3) “We have indeed computed the diamond nucleation rate from liquid carbon directly at a number of conditions.” “the nucleation rate $J = k \cdot \exp[-G^*/kBT]$ can be obtained without any assumption from classical nucleation theory or linear fitting.” No, they did not. [...] The expression $J = k \cdot \exp[-G^*/kBT]$ is the typical Arrhenius relation from transition state theory. [...]

Here, I agree with the authors. The Arrhenius-like expression does not subsume CNT, so in this respect I am fine with how the nucleation study was carried out. However, see my previous comment on considering a better description of the crystallite (more/better Cvs).

- (4) “We have indeed applied the Mermin finite-temperature formalism. It is not correct that “the temperature of electrons in this study is always 0 K”.
- [...] The authors used the PBE functional, which was designed at 0 K, so in this work, the exchange-correlation interactions of electrons, which play a key role in DFT, were basically simulated at 0 K.

Here, I agree with the reviewer. The Fermi smearing is justified by the Mermin-functional formalism, but it is an approximation, which could be crude at extreme conditions. So, it is wrong to state that the Mermin finite-temperature formalism was applied. In the Fermi-smearing formalism, the electrons are moving on the zero-K PES, only their occupation is determined by the temperature of the system. So, changes in the PES determined by the coupled electron-ion motion are not captured. It is hard, I believe, to assess how much this approximation affects the results (note: this is on the ground-truth, i.e., DFT side; the MLP would very probably adapt if one had real finite-T-

electrons data). As mentioned in my previous reports, I just suggest that this approximation aspect is explicitly mentioned in the manuscript.

In summary, I essentially agree with Ref. #3 that the authors are overselling their study, as only thermodynamic aspects are taken care of, for the CH mixture, which would be the novelty of the paper. Otherwise, the work is extremely well carried out and the level of detailed information provided, both in the manuscript and SI should be a role model for similar *ab initio* + ML studies.

We thank the adjudicating referee for their careful reading of the exchange and for their fair assessment. We have made a number of changes to the manuscript and in the SI in response to their comments, and have highlighted these in blue in the revised version of the text. In what follows, we respond to each of the points raised by the referees. For clarity, Referee 3's comments and our previous response are in black, Referee 4's comments are in green, and our latest response is in blue.

Most importantly, we now focus the paper on the thermodynamics of diamond formation, as our key findings with planetary implications (i.e. phase boundaries of diamond formation, liquid-liquid phase separation, and the depletion zone) only rely on the thermodynamic analysis but not the kinetics. We removed the nucleation part and made it clear that we do not calculate the nucleation kinetics in the C-H system, starting from the title and throughout the entire paper. The specific changes are detailed below.

For this round of review, I will focus on the disagreements between the authors and referee #3, as asked by the editor. In particular, I will quote (excerpts from) and comment the document CH3-revision-3 (383671_3_art_file_7121085_rlw12x.pdf within NCOMM's reviewing system). Quoted ref. #3 comments are in black and authors' reply in blue. My comments are in green.

- Ref #3 - The title of this manuscript is "Diamond formation from hydrocarbon mixtures in planets". How come that "diamond formation from C/H mixtures" is beyond the scope?

My comment - The reviewer has a fair point here. I agree the title is overselling the results, given that only thermodynamics conditions are examined for CH mixtures, i.e., kinetic effects are not analysed. I strongly suggest this is made clear by modifying the title into "Thermodynamics conditions for ..." or something similar. I appreciate that the abstract has been already modified in this direction in the latest version.

Authors:

We have followed the suggestion of the Referee and changed the title into "Thermodynamics of diamond formation from hydrocarbon mixtures in planets".

- Authors - The referee insisted on studying the nucleation of diamond from hydrocarbons, but we think this is beyond the scope of the current paper, particularly considering the breadth, the technical difficulties and the implications of the results already included in the manuscript.

I agree that a thorough kinetic study for CH mixtures is more than a simple addition to the manuscript, which has its merits with the current content. However, I see that there is an asymmetry in reporting the kinetic study only for the pure C, then leaving the reader on a "cliff hanger" on how the kinetic story ends. I understand the authors' argument about the comparison to a previous study on diamond nucleation with a different model potential, but wouldn't it make more sense to have a dedicate paper on the kinetics, for both pure C and CH mixtures?

This would also allow the possibility to look more carefully into collective variables. As Ref #3 hints at in their previous report, it has been shown in the past that size of the crystallite might not be always the best choice, or, in general one may need more than one CV, i.e., by adding info on the shape of the crystallite (to my knowledge, this was first shown by Moroni, Ten Wolde, Bolhuis in their 2005's PRL, but I am sure the authors know the fundamentals in the field).

Authors:

We thank the Referee for understanding that the manuscript has its merits and that a thorough kinetic study for C/H mixture is difficult. We also agree that a dedicated paper on the kinetics for both pure C and C/H mixtures is a good idea.

We have thus removed the content on the diamond nucleation from pure C in the current manuscript, also because it is not related to the main implications of the paper (phase boundaries of diamond formation, liquid-liquid phase separation, the depletion zone).

In the future we plan a thorough study on the kinetics of diamond nucleation. The choice of the CV there is likely to be crucial and needs fine-tuning.

- Moreover, planets have billions of years to evolve, so the thermodynamic picture of whether diamonds can form or not is perhaps more important than the transient mechanism of diamond nucleation.

This is an unscientific reply. As the authors know, nucleation rates in condensed matter may well span tens of orders of magnitude. One may get a nucleation rate that is much less than one event per planet per solar system, in a region where thermodynamics says the crystal is more stable. So, it is not true that thermodynamic conditions are more important than the transient mechanism. I would suggest the authors tell clearly that kinetics could matter, which was the whole point of Ghiringhelli, Valeriani, et al., 2007, I understand.

Authors:

In the manuscript we now state very clearly the possible influence from the kinetics:

- In Fig 1 we show the inferred threshold conditions with a diamond nucleation rate of $10^{-40} \text{ m}^{-3}\text{s}^{-1}$ from pure liquid carbon by Ghiringhelli et al. 2007. We further explain that undercooled liquids can remain metastable for a long time as solidification is initiated by a kinetically activated nucleation process.
- In the discussion section of the manuscript, we wrote: "Note that our boundaries are solely based on the thermodynamic criterion, but the kinetic nucleation rate may play a role particularly in shock-compression experiments. An amount of undercooling may be needed for diamond to nucleate from C/H mixtures within finite time, depending on the magnitude of the nucleation activation barrier. In homogeneous nucleation, the magnitude of the interfacial free energy contribution is crucial~\cite{Kelton1991, Ghiringhelli2007}."

“As shown below and in the SI, the MLP reproduces the bond lifetimes extremely well.” I do not know how the authors defined “extremely well”, but it is very obvious that the two figures on the left panel are not the same, even though they showed very crude comparison. For example, between 0.8 and 1.25 g/cc, the C-H bond life time predicted by MLPs is apparently longer than that from DFT.

I partially agree with the referee, here. The plots, currently Fig. S14 in SI seem to show some possibly significant discrepancy between DFT and the MLP. I think the chosen layout of the plot is not very readable, with the color map. I suggest to have 5x2 plots (one per temperature and for both CC and CH bond lifetimes) with density on the x-axes and the lifetimes on the y-axes (plots can be merged if still legible).

This said, lifetimes might be more relevant for kinetics than thermodynamics, so as a recommendation for the future work on kinetics, the MLP might need some improvement. In short, I believe the authors are a bit overselling their MLP. But I am also confident that for a thermodynamic study the discrepancies are not crucial. As mentioned in earlier reports, I encourage the authors to be more open on the shortcomings, in the manuscript itself. I applaud the thorough DFT-MLP comparison reported in SI, but I would point out that, e.g., dynamic properties are only fairly good at low T and rho (Fig. S13), that liquid methane is quite overstructured at low rho and higher T.

Authors:

We apologize for the original layout of Fig. S14 in SI which was not optimal.

We have now added the suggested 5x2 plots with density on the x-axes and the lifetimes on the y-axes to the SI. It can be seen that the MLP captures the C-C bond lifetimes extremely well, across all the temperatures and pressures. The C-H bond lifetimes are in general very short, on the order of 0.005 ps. The MLP simulations systematically slightly underestimate the C-H bond lifetimes, although the overall trend is consistent with the DFT MD and the absolute difference is less than 0.001 ps at all conditions. This difference may also come from the different setups in the MD simulations.

We have also added discussions about the possible shortcomings of the MLP:

“However, we would like to caution the limitations of the current MLP: The MLP is not applicable to gas phase hydrocarbons. For low-pressure carbon phases and diamond-graphite transitions, the current MLP has not been extensively tested, and we recommend users to employ the MLP from Ref.~\cite{rowe2020accurate}. Long-range Van der Waals interactions in liquid methane at low density may be important~\cite{veit2019equation} but are lacking in the current MLP. The

comparison between PBE and MLP for structure and dynamic properties such as equation of states, radial distribution functions, diffusivity, vibrational density of states, and bond lifetimes are provided in the SI. We recommend checking these comparisons before applying the MLP for a given C/H composition at certain conditions.”

- Their MLPs are poorly constructed and not reliable. [...] The RMSEs in either training or test sets are way too large.

The referee’s statements are too strong, in my opinion. The MLP was carefully trained – and tested! – and I believe that the errors are acceptable for the thermodynamic study

Authors:

We thank the Referee for recognizing the careful training and the thorough benchmark of the MLP.

- The referee mentioned the training errors of other systems (low-pressure water and methane). However, the training error is only meaningful when comparing it to the energy span of the system.

Here, I disagree with the authors. The relative error, or error compared to the span of values of the trained quantity is the typical “ML experts” figure of merit, as one wants to know how better is the trained model than just predicting the average value of the quantity for every new data point. For a thermodynamic study, I would say that the errors have to be projected into reliability of the phase boundaries, e.g., in terms of kelvins and gigapascals. For instance, how reliable are the boundaries of the “depletion zone” in Fig. 3G, in terms of K and GPas?

Authors:

Our previous response was perhaps not clear. What we were trying to say is that the absolute training errors of MLPs for different systems cannot be compared directly. This is because different systems (e.g. ambient-pressure water and high-P C/H mixtures) have very different energy scales, so it is more meaningful to consider the ratio of the MLP training errors to the standard deviations of training energies.

We agree that the uncertainties of the MLP on the phase boundaries are useful. In our previous studies on low-pressure and high-pressure water (<https://www.nature.com/articles/s41567-021-01334-9>), we were able to use free energy perturbation to remove the error of the MLP. In the current case, due to the different C/H ratios and the LLPS mechanism this is difficult to do. As we mentioned in the first revision, another way to gauge the statistical errors of ML models is to use the committee model (e.g. <https://aip.scitation.org/doi/full/10.1063/5.0036522>). This is essentially fitting multiple MLPs using different splits of training sets or random initialization, and looking at the spread of the computed physical quantities from MD simulations employing the different MLPs. For this study, we have constructed two versions of the MLPs (MLP-V1 and MLP-V2) using different training

sets, and the MLP-V2 has better accuracy for structures with the low C concentrations. The two MLPs have very similar predictions on the phase separations. The difference in the pressure and temperature is below 300 K and 50 GPa, although this likely overestimates the uncertainty still as MLP-V1 is less accurate at the low C concentrations. This attests the robustness of our conclusions on the PS.

• (3) “We have indeed computed the diamond nucleation rate from liquid carbon directly at a number of conditions.” “the nucleation rate $J=k*\exp[-G^*/kBT]$ can be obtained without any assumption from classical nucleation theory or linear fitting.” No, they did not. [...] The expression $J=k*\exp[-G^*/kBT]$ is the typical Arrhenius relation from transition state theory. [...]

Here, I agree with the authors. The Arrhenius-like expression does not subsume CNT, so in this respect I am fine with how the nucleation study was carried out. However, see my previous comment on considering a better description of the crystallite (more/better Cvs).

Authors:

We agree with the comments. As described before, we have now removed the nucleation part from the current manuscript and will focus on this in a future project.

• (4) “We have indeed applied the Mermin finite-temperature formalism. It is not correct that “the temperature of electrons in this study is always 0 K”. [...] The authors used the PBE functional, which was designed at 0 K, so in this work, the exchange-correlation interactions of electrons, which play a key role in DFT, were basically simulated at 0 K.

Here, I agree with the reviewer. The Fermi smearing is justified by the Mermin-functional formalism, but it is an approximation, which could be crude at extreme conditions. So, it is wrong to state that the Mermin finite-temperature formalism was applied. In the Fermi-smearing formalism, the electrons are moving on the zero-K PES, only their occupation is determined by the temperature of the system. So, changes in the PES determined by the coupled electron-ion motion are not captured. It is hard, I believe, to assess how much this approximation affects the results (note: this is on the ground-truth, i.e., DFT side; the MLP would very probably adapt if one had real finite-T-electrons data). As mentioned in my previous reports, I just suggest that this approximation aspect is explicitly mentioned in the manuscript.

Authors:

We have followed the suggestion of the Referee. We have added the following sentences to the Supporting Information (section 1B) to clarify the ground-truth used in this study:

In this study we use an approximation to finite-temperature DFT. The Kohn-Sham electrons are assigned a temperature through a Fermi smearing of the orbital occupation numbers while the exchange-correlation (XC) functional is set to the zero-K PBE functional. A finite-temperature extension of the PBE functional is not yet available and so the impact of this approximation is not known. However, in the case of the local density approximation

(LDA) the zero-K XC approximation was shown to lead to very small changes to the equation of state for temperatures at or below 10000 K (less than a 1 % pressure difference) compared to the temperature-dependent LDA XC functional for both hydrogen [Ramakrishna et al., PRB 101, 195129 (2020)] and carbon [Bonitz et al., Phys. Plasmas 27, 042710 (2020)].

We have also adjusted the wording in the Methods section to highlight the approximative nature of the DFT reference.

In summary, I essentially agree with Ref. #3 that the authors are overselling their study, as only thermodynamic aspects are taken care of, for the CH mixture, which would be the novelty of the paper. Otherwise, the work is extremely well carried out and the level of detailed information provided, both in the manuscript and SI should be a role model for similar ab initio + ML studies.

REVIEWER COMMENTS

Reviewer #4 (Remarks to the Author):

I am fully satisfied by the authors' replies to my comments and suggestions as well as the changes in the resubmitted version of the manuscript.

I am particularly fond of all the discussions on the limitations of the modelling approaches; this is very good scientific practice!

I strongly recommend the present manuscript for publication on Nature Communications.

Reviewer #4 (Remarks to the Author):

I am fully satisfied by the authors' replies to my comments and suggestions as well as the changes in the resubmitted version of the manuscript.

I am particularly fond of all the discussions on the limitations of the modelling approaches; this is very good scientific practice!

I strongly recommend the present manuscript for publication on Nature Communications.

Authors:

We thank the Referee for their positive assessment and encouraging comment.